



# On the representation of water reservoir storage and operations in large-scale hydrological models: implications on model parameterization and climate change impact assessments

Thanh Duc Dang[1], AFM Kamal Chowdhury[1], and Stefano Galelli[1]

[1]Pillar of Engineering Systems and Design, Singapore University of Technology and Design, Singapore 487372

**Correspondence:** Stefano Galelli (stefano_galelli@sutd.edu.sg)

**Abstract.** During the past decades, the increased impact of anthropogenic interventions on river basins has prompted hydrologists to develop various approaches for representing human-water interactions in large-scale hydrological and land surface models. The simulation of water reservoir storage and operations has received particular attention, owing to the ubiquitous presence of dams. Yet, little is known about (1) the effect of the representation of water reservoirs on the parameterization of

hydrological models, and, therefore, (2) the risks associated to potential flaws in the calibration process. To fill in this gap, we contribute a computational framework based on the Variable Infiltration Capacity (VIC) model and a Multi-Objective Evolutionary Algorithm, which we use to calibrate VIC's parameters. An important feature of our framework is a novel variant of VIC's routing module that allows us to simulate the storage dynamics of water reservoirs. Using the upper Mekong river basin as a case study, we calibrate two instances of VIC—with and without reservoirs. We show that both model instances

have the same accuracy in reproducing daily discharges (over the period 1996–2005); a result attained by the model without reservoirs by adopting a parameterization that compensates for the absence of these infrastructures. The first implication of this flawed parameter estimation stands in a poor representation of key hydrological processes, such as surface runoff, infiltration, and baseflow. To further demonstrate the risks associated to the use of such model, we carry out a climate change impact assessment (for the period 2050–2060), for which we use precipitation and temperature data retrieved from five Global

Circulation Models (GCMs) and two Representative Concentration Pathways (RCPs 4.5 and 8.5). Results show that the two model instances (with and without reservoirs) provide different projections of the minimum, maximum, and average monthly discharges. These results are consistent across both RCPs. Overall, our study reinforces the message about the correct representation of human-water interactions in large-scale hydrological models.

## 1  Introduction

Hydrological systems consist of multiple physical, chemical, and biological processes, most of which are profoundly altered by anthropogenic interventions (Nazemi and Wheater, 2015a, b). Land cover modifications or hydraulic infrastructures, for





instance, affect both surface and sub-surface hydrological processes by redistributing water over time and space (Haddeland et al., 2006; Bierkens, 2015). Such alterations are expected to amplify in the near future, owing to the increase in water and energy consumption (Abbaspour et al., 2015). In this context, hydrological models play a key role, as they help plan the use of water resources in a sustainable way, so as to avoid adverse impacts on ecosystems and livelihoods (Bunn and Arthington, 2002; Yassin et al., 2019). A detailed and accurate representation of the anthropogenic interventions within hydrologic models is thus of paramount importance: successful water management plans must necessarily build on reliable models.

Water reservoirs are arguably one of the most common infrastructures altering hydrological processes at the catchment scale; yet, their representation in hydrological and land surface models is challenged by multiple factors. First, the vast majority of the models currently available was initially conceived to study and understand the behaviour of natural systems, so the added representation of water reservoirs entails the partial modification of the model structure. Second, the existing databases (e.g., GRanD; Lehner et al. (2011)) provide details on dam design specifications, but no information on the management aspects, such as the operating rules or flood contingency plans. Third, the installation of dams is generally combined with impoundment (or filling) strategies, which may largely differ from the steady-state operating rules and last from a few months to multiple years (Gao et al., 2010; Zhang et al., 2016). Although the complexity of these factors varies with the study site at hand, one might imagine that the representation of water reservoir storage and operations is particularly challenging for large-scale models, simply because of the number of dams deployed over time in large river basins. It is perhaps not surprising to observe that water reservoirs—and their corresponding operations—have not been consistently accounted for across the broad number of large-scale hydrological modelling studies available in literature.

A simple and popular approach stands in the exclusion of large impounds from the streamflow routing modules; a modelling choice that has been adopted in many regions across the globe (Maurer et al., 2002; Jayawardena and Mahanama, 2002; Akter and Babel, 2012; de Paiva et al., 2013; Leng et al., 2016). Such approach can support the investigation of various physical processes (e.g., emergence of new hydrological regimes, generation of land surface fluxes), but obviously prevents the application of the hydrological models to downstream water management problems, such as investigating the impact of regime shifts on hydropower production. Another potential issue with this approach stands in the model parameterization, which might be affected by a calibration process carried out with hydrological time series altered by anthropogenic interventions. de Paiva et al. (2013), for instance, implemented the MGB-IPH hydrologic/hydraulic model to the Amazon River basin—a region characterized by the presence of hydroelectric dams (Finer and Jenkins, 2012)—and yet obtained reliable calibration performance at multiple gauging stations. A similar example is represented by Abbaspour et al. (2015), who simulated hydrological and water quality processes for the entire European continent. Despite neglecting the presence of hydraulic infrastructures, the model yielded acceptable values for the goodness of fit statistics. One may thus wonder whether the calibration process somehow compensates for a deficiency in the model structure.

With the goal of striking a balance between an accurate representation of reservoirs and the 'costs' due to the modification of the model structure, several researchers have adopted an hybrid approach, in which the output of hydrologic/hydraulic models (e.g., runoff or streamflow at multiple locations) is post-processed with the aid of water management (or reservoir operation) models. The very first efforts employed data on water uses to correct the output of global models, such as WaterGAP (Alcamo





et al., 1997) or WBM (Vörösmarty et al., 1998). Using a similar concept, Hanasaki et al. (2006) accounted for 452 reservoirs in a global river routing model. More sophisticated post-processing techniques are based on optimization algorithms, which are used to design either reservoir operating rules or sequences of reservoir discharges that meet pre-defined objectives (e.g., hydropower production). Lauri et al. (2012) and Hoang et al. (2019), for example, first calibrated the distributed hydrological

model VMod for the Mekong river basin, and then post-processed its output using a Linear Programming algorithm that designed the discharge time series for 126 dams over a given simulation scenario. Similarly, Turner et al. (2017) and Ng et al. (2017) examined the vulnerability of global hydropower production to climate changes and El Niño Southern Oscillation by correcting the discharge simulated by WaterGAP. In this case, the correction entailed designing bespoke reservoir operating rules through the use of a Stochastic Dynamic Programming algorithm (Turner and Galelli, 2016). Other recent applications of

post-processing techniques were adopted in Masaki et al. (2017); Veldkamp et al. (2018); Zhou et al. (2018).

Naturally, the most suitable approach stands in the direct representation of water storage and operations within a large-scale hydrological model (Bellin et al., 2016). This approach requires not only to modify the model structure (or to develop a new one), but also to gather information on the design specifications and operating rules of the water reservoirs. Because of these challenges, the number of large-scale hydrological modelling studies adopting such approach is limited. A first attempt was

carried out by Pokhrel et al. (2012), who incorporated a water regulation module into the MATSIRO model to reproduce the dynamics of heavily regulated global river basins. More recently, Shin et al. (2019) integrated a reservoir storage dynamics and release scheme into the continental hydrological model LEAF-Hydro-Flood to simulate ∼1,900 reservoirs within the contiguous United States. In both studies, the authors gave particular emphasis to the calibration of the reservoir operating scheme, and demonstrated that the hydrological model accurately represents some processes altered by human interventions,

such as the reservoir-floodplain inundation.

While the relevance and needs for the description of human-water interactions in hydrological models are now well acknowledged (Nazemi and Wheater, 2015a), less is know about the risks associated to a poor representation of such interactions. For example, can the estimation of some hydrological parameters be flawed by an inaccurate representation of water reservoir storage? What are the implications for the downstream applications of a flawed model? To answer these questions, we take the

upper Mekong river basin as a case study, for which we develop a computational framework based on the Variable Infiltration Capacity (VIC) model (Liang et al., 1994) and a Multi-Objective Evolutionary Algorithm (MOEA) tasked with the problem of calibrating the model. A key feature of the framework is a novel variant of VIC that allows us to represent the reservoir storage dynamics and operating rules within the streamflow routing module. In a first experiment, we use this framework to calibrate two instances of VIC—with and without reservoirs. As we shall see, both model instances attain the same accuracy; a result

obtained by the model instance without reservoirs by adopting a parameterization that compensates for the absence of these infrastructures. In turn, this leads to a poor representation of key hydrological processes, such as infiltration or baseflow. In our second experiment, we demonstrate the potential implications of these unintended consequences by applying two selected model instances (with and without reservoirs) to a climate change impact assessment, for which we obtain partially-diverging expectations on the hydrological alterations caused by global warming.





In the remainder of the manuscript, we first describe the study area (Section 2) and then proceed by illustrating the computational framework (Section 3), including the data on dams and operating rules. In Section 4, we provide a detailed description of the results obtained for the aforementioned experiments, whose implications are further discussed in Section 5.

## 2   Study area

The Mekong is a trans-boundary river that flows through China, Myanmar, Thailand, and Laos before pouring into one of the world's largest delta located in Cambodia and Vietnam. The catchment area of about 795,000 km$^2$ can be divided into two parts, namely the upper Mekong, or Lancang, and the lower Mekong basins (Figure 1a). The upper Mekong stretches in a North-to-South direction, and is characterized by a complex orography, with high mountains and deep valleys (Figure 1b). Because of these orographical conditions, the spatio-temporal variability of rainfall and temperature is remarkable. The average

annual precipitation across the basin ranges from 752 to 1,025 mm, 70% of which is concentrated in the monsoon season (May to November). The precipitation in the Northwestern part of the basin is sometimes lower than 250 mm/year, making it dryer than the Southeastern part, which receives an average of 1,600 mm/year (Han et al., 2019). The average annual temperature across the basin varies narrowly (from 12.3 to 14.3 °C), but the latitudinal temporal gradient is much larger—about 2.2 °C/100 km (Wang et al., 2014). Climate changes are expected to modify both rainfall and temperature patterns, making the region

warmer, wetter, and more susceptible to extreme weather events (Tang et al., 2015).

The favourable orography and abundant water availability have attracted massive investments in the hydropower sector (see the location of the dams in Figure 1b), with consequent impacts on the riverine ecosystems (Lauri et al., 2012; Dang et al., 2018; Hoang et al., 2019). The impact of these dams goes beyond the upper Mekong basin (Zhao et al., 2012; Han et al., 2019): the analysis of historical data shows that dams have already modified many indicators of hydrological alterations in the entire

basin, including the Cambodian lowlands and the river delta (Hecht et al., 2018). These alterations appear to be more evident since 1992, when the Manwan dam started storing water (Cochrane et al., 2014; Lu et al., 2014; Dang et al., 2016). Overall, the upper Mekong basin offers two desirable features for investigating the effect of water reservoir storage and operations on the parameterization of hydrological models. First, the catchment is heavily regulated (Hecht et al., 2018). Second, the catchment area is about 24% of the whole Mekong River basin, so this helps reduce the computational requirements of the optimization-

based calibration process. The location of the gauging station used in our work (Chiang Saen) is illustrated in Figure 1a. This station provides a long and reliable streamflow time series, which has been adopted by several studies on the Mekong basin (e.g., Lauri et al. (2012); Cochrane et al. (2014); Lauri et al. (2014); Hoang et al. (2016)). In this study, we use daily discharges measured in the period 1996–2005 for the model calibration.

## 3   Materials and methods

The first goal of our study is to investigate the role of water reservoir storage and operations on the parameterization of large-scale hydrological models. To this purpose, we adopt the computational framework illustrated in Figure 2, which consists of





VIC's rainfall-runoff and routing modules and the $\varepsilon-$NSGAII MOEA. In Section 3.1 we provide a detailed description of VIC's modules, including the proposed variant for representing reservoir storage dynamics. The data and experimental setup of the framework are outlined in Section 3.2 and 3.3. In Section 3.4, we describe the climate change data used for our second goal, that is, to demonstrate that different model parameterizations caused by the absence (presence) of water reservoirs can

affect the results of a climate change impact assessment.

## 3.1 Hydrological-water resources management model

### 3.1.1 Variable Infiltration Capacity model

VIC is a large-scale, semi-distributed land hydrological model maintained and developed by the University of Washington (http://www.hydro.washington.edu). The model consists of two core components, namely a rainfall-runoff and routing module

(Figure 2), which can be applied to multiple spatial scales and implemented with different temporal resolutions—daily, in our case. The rainfall-runoff module simulates the water and energy fluxes that govern the terrestrial hydrological cycle (Liang et al., 1994). To this purposes, it takes as input climate forcings (precipitation and temperature), land use and soil maps, Leaf Area Index and albedo, and a Digital Elevation Model (DEM). For each computational cell, the module uses one vegetation layer and two (or three) soil layers: the upper soil layer controls evaporation, infiltration, and runoff, while the lower layer

controls the baseflow generation. These gridded variables are then used by the routing module (Lohmann et al., 1996, 1998), which calculates the streamflow using the Unit Hydrograph approach and linearized de Saint-Venant equations.

Following the approach adopted in previous works on the calibration of VIC (e.g., Dan et al. (2012); Park and Markus (2014); Xue et al. (2015)), we focus our attention on six main parameters that control the rainfall-runoff process (Table 1). These parameters are the thickness of the two soil layers ($d_1$ and $d_2$), the infiltration parameter b, and three baseflow parameters ($D_s$,

$D_{max}$, and $W_s$). The parameter b characterizes the shape of the Variable Infiltration Capacity curve, and therefore influences the available infiltration capacity and the quantity of runoff generated by each cell (for additional details, please refer to Ren-Jun (1992) and Todini (1996)). A higher value of b leads to a lower infiltration rate and higher surface runoff. The three parameters $D_s$, $D_{max}$, and $W_s$ determine the shape of the Baseflow curve (Franchini and Pacciani, 1991), which relates the soil moisture in the lower layer to the amount of baseflow. More specifically, $D_{max}$ is the maximum baseflow that can occur in the lower layer,

while $D_s$ is the fraction of $D_{max}$ associated to the transition from linear to non-linear (rapidly increasing) baseflow generation. $W_s$ is the fraction of the maximum soil moisture (in the lower layer) where non-linear baseflow occurs. Hence, higher values of $W_s$ increase the water content needed for rapidly increasing baseflow. The thickness of the two soil layers affects several processes. In general, thicker layers delay the seasonal peak flow and increase the evaporation losses (since they increase the water storage capacity).

### 3.1.2 Water reservoir storage and operations

To represent the storage dynamics of water reservoirs, we modified VIC's routing module (version 4.2) using the following steps. First, we determine the location of all dams within the basin, and directly add them to the model using a dam cell



(Figure 3a-b). To avoid allocating multiple dams within the same cell, we adopt a high-spatial resolution of 0.0625 degree (approximately 6.9 km). Then, we aggregate the reservoir storage in the dam cell, from which water is discharged using the rule curves described in the following paragraph. Since the construction of a dam is likely to create an impoundment with surface area larger than the dam cell, we proceed by estimating the maximum reservoir extent; an information used to

determine the so-called reservoir cells, namely cells that are at least half-covered by water (see Figure 3b). Although these cells do not contain the reservoir storage, they can affect the evaporation processes, so their number and location must be determined accurately. The flow routing in these cells follows the information provided in the flow direction map (described in Section 3.2.1). We note that a more realistic way of representing a reservoir within a hydrological model is to spread the reservoir storage over multiple upstream cells from the dam location (Shin et al., 2019). Yet, a successful implementation of this method

would require a detailed bathymetry of all reservoirs within the basin (an information that may not always be available) and a 2D model of the reservoir, so as to accurately calculate the water fluxes between the different reservoir cells.

    As for the reservoir operations, we adopt an approach similar to that of Piman et al. (2012), which relies on rule curves conceived to maximize the hydropower production—an assumption justified by the fact that all dams within the upper Mekong are operated for hydropower supply (Räsänen et al., 2017). Determining the rule curve for a given reservoir means determining

the daily target water levels. For the case of hydropower production in the Mekong basin, such rule should allow to (1) drawdown the reservoir storage during the drier months (e.g., December to May) to maximize the production of electricity, (2) recharge the depleted storage during the monsoon season, and (3) avoid the risks of spilling water at the end of the monsoon season (see the illustration in Figure 3c). Such rule can be tailored to each reservoir within the basin by determining the time at which the minimum and maximum water levels are reached (May and November, in the Mekong; Piman et al. (2012)), and

setting the value of the minimum and maximum water levels. In our case, we use the minimum and maximum elevation levels of each reservoir.

    As shown in Figure 3c, there are three water levels that divide the storage into four zones. These levels are the dead water (or minimum elevation) level, the target water level, and the full (or maximum elevation) level. If the water level falls below the dead water level (Zone 1), the turbines are not operated. If the level is between the dead water and target level (Zone 2),

the model first uses the information on the incoming daily inflow to solve a mass balance equation, in which the discharge from the dam is kept at zero. This is aimed to understand whether the water level is expected to go beyond the target at the end of the day. If that is the case, the model discharges through the turbines the amount of water needed to keep the level close to the target. Otherwise, the turbines are not activated. In Zone 3 (between the target and full level), the turbines are used at their maximum capacity, until the water reaches the target level. In Zone 4 (i.e., level above the maximum elevation), both

turbines and spillways are used. The key advantage of the rule curves adopted here is that they do not require the calibration of any parameter. Naturally, such approach is less applicable when the information on the reservoir operating objectives is not available, or when dealing with multi-purpose water systems.


## 3.2 Data and preprocessing

### 3.2.1 Climate forcings and other input variables

Climate forcings are represented by precipitation and air temperature (maximum and minimum), which must be provided at a daily time step. As far as precipitation is concerned, we use the APHRODITE dataset (Asian Precipitation - Highly-Resolved

Observational Data Integration Towards Evaluation), developed by the University of Tsukuba, Japan, using rain-gauge data (Yatagai et al., 2012). APHRODITE is available with a spatial resolution of 0.25 degree, and has been shown by Lauri et al. (2014) to be the most suitable precipitation dataset available for the Mekong basin. A similar observation applies to the CFSR (Climate Forecast System Reanalysis) maximum and minimum temperature dataset (Saha et al., 2014). These data are then interpolated to meet the spatial resolution of 0.0625 degrees adopted in our VIC model implementation. More specifically, we

use the bilinear interpolation method, which has found successful application in some recent studies (e.g., Hoang et al. (2016); Shin et al. (2019)). We also bias correct the APHRODITE dataset (using a multiplying factor of 1.26), as recommended by Lauri et al. (2014).

Land use and land cover data are obtained from the Global Land Cover Characterization (GLCC) dataset, developed by the United States Geological Survey. We choose this product because it was completed in 1993, close to the simulation period

adopted in our study (1995–2005). With such choice, we make sure that the influence of land use dynamics on the model parameterization is minimized. Soil data are extracted from the Harmonized World Soil Database (HWSD), developed by the International Institute for Applied System Analysis and Food and Agriculture Organization, and last updated in 2013. Both land use and soil maps are generated with the majority resampling technique, since their original spatial resolution is 30 arcsecond (approximately 1 km). This technique assigns the most common values found from the group of involved pixels to the new

cell. The resulting maps are illustrated in Figure 4a-b. The monthly Leaf Area Index and albedo are derived from the Moderate Resolution Imaging Spectroradiometer (Terra MODIS) satellite images, which represent changes in canopy and snow coverage over time. (It is worth noting that snowmelt only marginally contributes to the streamflow of the Mekong River; Räsänen et al. (2016).)

To estimate the flow directions, we use the Global 30 Arc-Second Elevation (GTOPO30) DEM, which has been adopted

in several studies (e.g., Kite (2001); Wu et al. (2012); Li et al. (2013)). First, we mask this DEM with the shape of the upper Mekong basin. Since GTOPO30 has a spatial resolution of 30 arcsecond, we then resample the DEM to the resolution of our VIC model using the average resampling technique (Hoang et al., 2019). Finally, we manually correct the flow direction map generated by ArcGIS by comparing it to a detailed river network provided by the Mekong River Commission. Such correction is necessary, since errors are to some extent unavoidable when automatically generating a flow direction map—because overland

runoff and interflow directions depend on the relation between hillslope characteristics and adopted spatial resolution. The resulting flow direction map is illustrated in Figure 4c.





### 3.2.2 Dams and reservoir informations

Our model requires detailed information on the reservoirs, namely location, storage capacity, dam height, dead storage, turbine design discharge, and maximum and minimum elevation levels. Such information (summarized in Table 2) was retrieved by cross-checking the databases provided by the Mekong River Commission, the International Commission On Large Dams,

and the Global Reservoir and Dam Database. Since data on the bathymetry of the reservoirs are not available, we modelled the storage-depth relationship with Liebe's method, which assumes that the reservoir is shaped like a top-down pyramid cut diagonally in half (Liebe et al., 2005). In other words, the relation between reservoir volume ($V$) and depth (or level, $h$) is equal to $V = ah^3$, where $a$ is a shape factor equal to $V_{cap}/h_{max}^3$ ($V_{cap}$ is the live storage capacity and $h_{max}$ the maximum water depth). This method has been adopted for regional and global studies (see Ng et al. (2017); Shin et al. (2019)).

As for the maximum reservoir extent (needed to determine the reservoir cells), the existing databases do not provide detailed information, such as the reservoir polygon, so we proceeded by analyzing remote sensed data. More specifically, we extracted surface water profiles from Landsat TM and ETM+ imagery. Landsat images are raster grids with seven layers corresponding to seven bands (excluding the panchromatic band). The Normalized Difference Water Index (NDWI) was calculated using the near-infrared (NIR, Band 4) and Short-Wave infrared (SWIR, Band 5) bands: NDWI=(NIR-SWIR)/(NIR+SWIR). Water

bodies have NDWI values greater than 0.3 (McFeeters, 2013), so from the NDWI raster we can create a binary raster in which 1 denotes a reservoir cell (and 0 a non-reservoir cell). This process can yield an accurate estimation of the reservoir cells, since Landsat images have a spatial resolution of 30 x 30 m.

### 3.3 Experimental setup

To carry out the calibration exercise (with and without reservoirs), we couple VIC with the $\varepsilon-$NSGAII algorithm (Reed et al.,

2013), which has found successful application in many water resources problems—including model calibration (ibidem). In our case, the decision variables are represented by the six parameters controlling the rainfall-runoff process in VIC (Section 3.1.1), and whose range of variability is reported in Table 1. As for the objective functions, we consider two goodness of fit statistics dependant upon the simulated streamflow, namely the Nash-Sutcliffe Efficiency (NSE) and Transformed Root Mean Square Error (TRMSE), which assess the model performance on high and low flows, respectively (Dawson et al., 2007). The

NSE is defined as:

$$NSE = 1 - \frac{\sum_{t=1}^{n}(Q_s^t - Q_o^t)^2}{\sum_{t=1}^{n}(Q_o^t - \overline{Q_o})^2}, \tag{1}$$

where $n$ is the number of time steps, $Q_s^t$ the simulated streamflow (at time $t$), $Q_o^t$ the observed streamflow (at Chiang Saen station), and $\overline{Q_o}$ the mean of the observed streamflow. The TRMSE is defined as:

$$TRMSE = \sqrt{\frac{1}{n}\sum_{t=1}^{n}(z_{s,t} - z_{o,t})^2}, \tag{2}$$

where $z_{s,t}$ and $z_{o,t}$ represent the value of the simulated and observed streamflow (at time $t$) transformed by the expression $z = \frac{(1+Q)^\lambda - 1}{\lambda}, (\lambda = 0.3)$. In other words, $\lambda$ scales down the values of the streamflow, and TRMSE thus emphasizes the errors





on the low flows. In this specific modelling problem, capturing both high and low flows is particularly important, since the riverine ecosystems are sensitive to both dry and wet conditions (Hoang et al., 2016).

Both objective functions are calculated for the period 1996–2005—after a one-year spin-up period, 1995—and scaled between 0 and 1, so we set only one value of $\varepsilon$ (equal to 0.001). The other $\varepsilon-$NSGAII parameters to setup are the size of the initial

population and the number of function evaluations, which are equal to 10 and 250—a setting that strikes a reasonable balance between the computational requirements of the calibration exercise and the quality of the solutions. Each calibration exercise (with and without reservoirs) is solved with 20 different random seeds, so as to characterize the variability in the $\varepsilon-$NSGAII stochastic search process. The final set of Pareto-efficient solutions (i.e., alternative parameterizations of VIC) thus corresponds to the set of Pareto-efficient solutions identified across all 20 seeds. All experiments are carried out on an Intel (R) Xeon (R)

W-2175 CPU 2.50 GHz with 128 GB RAM running Linux Ubuntu 16.04 (Xenial Xerus), using a Python implementation of various MOEAs (Platypus) that allows to parellelize the optimization experiments. The average runtime (across the 20 seeds) is about 200 hours.

Since six (out of eleven) dams became operational during the study period (see Table 2), the VIC simulation with reservoirs is implemented in such a way to activate the reservoirs at the right time. In this specific implementation, we do not use filling

strategies different from the rule curves described in Section 3.2.2, because all six dams reach a steady-state operation within a few months (data not shown).

## 3.4 Climate change data

For our second experiment, we used the CMIP5 climate projections to derive climate change scenarios for the period 2050–2060. Since the data provided by the Coordinated Regional Climate Downscaling Experiment only cover one GCM for our

study site (Giorgi and Gutowski Jr, 2015), we followed the approach taken by previous studies (e.g., Hoang et al. (2016, 2019)) and proceeded by using GCM projections as basis for our scenarios. As far as the GCMs are concerned, we used ACCESS1-0, CCSM4, CSIRO Mk3.6, HadGEM2-ES, and MPI-ESM-LR, whose reliability for this region has been evaluated in a few previous studies (Sillmann et al., 2013; Huang et al., 2014; Ul Hasson et al., 2016; Hoang et al., 2016). The main characteristics of the GCMs are summarized in Table 3. As for the Representative Concentration Pathways (RCPs), we chose RCPs 4.5 and

8.5. The former is a medium-to-low scenario that assumes a stabilization of radiative forcing to 4.5 W m$^{-2}$ by 2100, while the latter is a high emission scenario based on an increase of the radiative forcing to 8.5 W m$^{-2}$ by 2100. These two RCPs should provide a broad range of climate variability for the region—and thus exclude RCP 2.6, which is characterized by the lowest radiative forcings.

To prepare the precipitation and temperature data used by VIC, we then re-gridded and bias-corrected the GCMs outputs.

The first step is necessary to overcome the limited spatial resolution of the GCMs (our VIC implementation uses a resolution of $0.0625° \times 0.0625°$), and is carried with the bilinear interpolation method. The bias-correction is performed with the delta method (Diaz-Nieto and Wilby, 2005; Choi et al., 2009), which has already been applied to our study site (Lauri et al., 2012).





With this method, we calculate correcting factors for precipitation and temperature using the following expressions:

$$\Delta_{PRE} = \frac{\bar{P}_{\text{series},i}}{\bar{P}_{\text{ref},i}}, \tag{3}$$

$$\Delta_{TEMP} = \frac{\bar{T}_{\text{series},i} - \bar{T}_{\text{ref},i}}{\sigma_{\text{ref},i}}, \tag{4}$$

where $\bar{P}_{\text{series},i}$ and $\bar{T}_{\text{series},i}$ are the (11 year) average precipitation and temperature for month $i$ produced by the GCM in our control period (1995–2005), $\bar{P}_{\text{ref},i}$ and $\bar{T}_{\text{ref},i}$ the (11 year) average observed precipitation and temperature for month $i$ in the

period 1995–2005, and $\sigma_{\text{ref},i}$ the standard deviation of the monthly average temperature during the same period for month $i$. These factors were then used to correct the future climate projections for each time series (using the same factor for all daily data in a given month).

The impact of climate change on hydrological processes are often assessed by studying changes in the flow regime, and, in particular, changes in the monthly, seasonal, and annual river discharges (Lauri et al., 2012, 2014). More recently, some

studies have focussed on hydrological extremes, such as high ($Q_5$) and low flows ($Q_{95}$) (Hoang et al., 2016). Since our goal is to demonstrate that different model parameterizations caused by the absence (presence) of water reservoirs can largely impact the results of climate change assessments—and not to push forward the boundaries of climate change impact assessments—we chose a simple and established criterion, namely the annual and monthly river discharges at the catchment outlet (Chiang Saen gauging station).

# 4 Results

To discuss about the impact of water reservoirs on the parameterization of hydrological models, we first compare the results of the calibration exercise carried out with and without reservoirs, and then proceed by comparing the performance of two selected parameterizations on the climate change impact assessment.

## 4.1 Model parameterization

The optimization-based parameterization exercise yielded a total of 118 and 109 parameterizations (or Pareto-efficient solutions) for the VIC implementations with and without reservoirs, respectively. To prove our hypothesis that the calibration process may somehow compensate for a deficiency in the model structure—the absence of reservoirs, in our case—we begin by analyzing the values of the goodness of fit statistics, namely NSE and TRMSE. Figure 5 reports the probability plots of NSE and TRMSE values obtained for the two model setups: results show that the calibration exercise yields a reasonable mod-

elling accuracy, with NSE and TRMSE varying in the ranges 0.68–0.79 and 8.10–16.69. More interestingly, these results show that the NSE and TRMSE values of both model setups belong to the same range of variability and follow an almost identical theoretical distribution (red and blue dashed lines). In addition, all NSE and TRMSE values of the models without reservoirs fall within the 95% confidence limits calculated using the NSE and TRMSE values attained by the models with reservoirs. To





corroborate this finding, we carried out a Kolmogorov-Smirnov two sample test to reject the null hypothesis that the values of NSE (and TRMSE) produced by the two model setups come from the same distribution. For both goodness of fit statistics, the hypothesis cannot be rejected (with a 5% significance level). Overall, this confirms that the accuracy of the models is not affected by the presence (absence) of the reservoirs.

How does the parameterization compensate for the absence of water reservoirs? To answer this question, we visualize both goodness of fit statistics (NSE and TRMSE) and model parameters ($D_s$, $D_{max}$, $W_s$, b, $d_1$ and $d_2$) in a parallel-coordinate plot (Figure 6). These eight variables are shown in eight parallel axes, so each line connecting the axes represents a parameterization (i.e., a solution of the optimization problem) along with the corresponding value of the goodness of fit statistics (i.e., the objectives). Blue and red lines denote solutions obtained with and without reservoirs, respectively. First of all, one can notice

that while NSE and TRMSE spread over the same ranges (results discussed in the previous paragraph), the presence/absence of reservoirs consistently yields different parameterizations. Let's analyze them. The value of b—characterizing the shape of the Variable Infiltration Capacity curve—belongs to two distinct ranges (0.319–0.495 and 0.002–0.195) for the model implementation with and without reservoirs, respectively, indicating that the model without reservoirs has higher infiltration and lower surface runoff than the model with reservoir (recall that a higher value of b leads to a lower infiltration rate and higher surface runoff; Section 3.1.1). A similar observation applies to the parameters $D_s$, $D_{max}$, and $W_s$, which determine the

shape of the Baseflow curve. In this case, the model without reservoirs has higher values of $D_{max}$ (i.e., maximum baseflow) and lower values of $D_s$ and $W_s$ (i.e., fraction of $D_{max}$ where rapidly increasing baseflow begins, and fraction of the maximum soil moisture in the lower layer where rapidly increasing baseflow occurs), suggesting that the absence of reservoirs leads to model paramaterizations that favour the generation of baseflow in the lower layer. Finally, we can note that $d_1$ (the thickness

of the first layer) in the models without reservoirs tends to be larger, indicating that these model instances increase the water storage capacity of the top layer. Overall, it appears that the calibration process compensates for the absence of water reservoirs by determining values of the soil parameters that can somehow 'mimic' the alterations caused by water reservoirs, namely an increase in the evaporation and delay in the peak flows—obtained by increasing infiltration, baseflow, and soil water storage capacity.

To further understand the unintended consequences of the absence of water reservoirs, we select two model parameterizations (with and without reservoirs) characterized by the same performance over the period 1996–2005. The values of NSE, TRMSE, and model parameters are illustrated in Figure 7a, while the simulated daily discharges produced by both models are compared in the scatter plot of Figure 7b. (The two models were also tested on a independent validation period, 1985–1995, for which we obtained a value of NSE equal to 0.724 (with reservoirs) and 0.718 (without reservoirs), and a value of TRMSE of 11.735

and 12.103.) In Figure 8, we contrast the average values of simulated baseflow and runoff during the dry (December–April) and wet (May–November) seasons of the period 1996–2005. Unsurprisingly, results show that during the dry season the model without reservoirs generates more baseflow and runoff than the model with reservoirs (left four panels of Figure 8): during the dry months, hydropower reservoirs release part of the water stored during the monsoon (recall the rule curves described in Section 3.1.2); a process simulated by the model without reservoirs by increasing both baseflow and runoff—and, therefore,

the discharge at the catchment outlet. During the wet season, we find an opposite trend: in these months, hydropower reservoirs





tend to store part of the water (thus reducing the discharge at the catchment outlet), so the model without reservoirs slightly decreases the discharge by reducing baseflow and runoff (right four panels of Figure 8). We also note that the difference between the two models is clearer during the dry season, when a larger amount of the water volumes is controlled by the hydropower reservoirs. One may thus suspect that these unintended consequences could further propagate in a downstream

application of the models, such as a climate change impact assessment.

## 4.2   Climate change impact assessment

To begin the climate change impact assessment, we compare the data produced by the GCMs over the reference and future period (1996–2005 and 2050–2060). In general, the total annual precipitation in the Lancang basin is projected to increase under almost all climate change scenarios—only the CSIRO MK3-RCP 8.5 scenario projects a -3.12% decrease in the total

annual precipitation. Yet, we observe a large spatial variability in the total annual rainfall within each scenario (see Figure 9). For example, in ACCESS-RCP 4.5, rainfall changes vary between -2% in the central part of the basin to more than +10% in the southern part. All scenarios (but for CSIRO MK3-RCP 8.5) tend to share a similar spatial pattern, in which the lower part of the basin exhibits an increase in the projected precipitation. As for the temperature, we observe an increase in both minimum and maximum temperature across all scenarios (see Figure 10), with higher warming for the RCP 8.5. Also in this case, we can note

some variability across the GCMs as well as the spatial domain. As discussed in Hoang et al. (2016), these precipitation and temperature scenarios represent an improvement with respect to the CMIP3 ones, which shown a broader variability. However, there still are some non-negligible differences across the scenarios that are likely to cause different projections of the annual and monthly river discharges.

The expected impact of climate changes on the annual river discharges is synthesized in Table 4, where we report the relative

changes in discharge with respect to the period 1996–2005. These results show an expected increase in the river flow under climate change. The ensemble mean for RCP 4.5 and RCP 8.5 is rather similar (+13.56 and +13.83% for the model with reservoirs, +13.62 and +13.92% for the model without reservoirs), while the range of variability is different. All scenarios under RCP 4.5 are associated to an increase in the discharge, while the scenario predicted by the CSIRO GCM (under the RCP 8.5) is associated to a slight decrease in the river discharge. Overall, these results do not highlight any macroscopic difference

between the projections issued by the two VIC implementations.

Figure 11a-b illustrates the projected monthly discharges at Chiang Saen for RCP 4.5 and RCP 8.5 (for the model without reservoirs). First of all, we can note that the ensemble range is broader during the monsoon months. This is an expected result, since the flow regime of the Lancang is driven by the monsoon. Results also show that the majority of the scenarios cause higher discharges (with respect to the baseline) during most of the months. This result is simply explained by the expected

increase in precipitation described above. In terms of RCPs, it appears that the ensemble range of the RCP 8.5 is slightly broader than the RCP 4.5's one, and that RCP 8.5 is expected to cause higher flows during the months of September, October, and November.

What is perhaps more interesting is a comparison between the projections yielded by the models with and without reservoirs. Both models produce similar ensemble ranges (see Figure 11a-d); yet, a closer analysis of the data reveals a difference in





the minimum, maximum, and average monthly discharges (across the GCM scenarios) produced by the two models (Figure 11e-f). In particular, the model with reservoirs predicts higher discharges in the July–September period and lower discharges in October and November. Note that such difference is consistent across both RCPs. Since both models share the same rainfall and temperature scenarios, the only cause for this stark difference can stand in the unintended consequences of the parameterization

process. As explained in Section 4.1, the model without reservoirs shows two 'artefacts' that help compensate for the absence of the hydropower reservoirs: first, it increases both baseflow and runoff during the dry season (to account for the water discharged to sustain hydropower production in the dry months); second, it decreases baseflow and runoff (to account for part of the water stored by the dams during the wet months). The latter artefacts is responsible for the macroscopic change in the hydrograph described above. In the wetter conditions depicted by the GCM-RCP scenarios, the hydropower reservoirs of the

Lancang basin receive larger inflows, part of which is directly spilled into the downstream reaches (data not shown). This is an unprecedented situation for the model without reservoirs, which cannot simulate an increase in the use of the spillways. In fact, this model tends to reproduce the dynamics learned during the calibration process, that is, storing part of the water (in the lower soil layer) during the monsoon season and slowly discharging it in the following months.

## 5   Discussion and Conclusions

This work contributes to the existing literature on large-scale hydrological modelling by studying the effect of water reservoir storage and operations on the parameterization of process-based models. To this purpose, we developed a computational framework consisting of VIC and the Multi-Objective Evolutionary Algorithm $\varepsilon-$NSGAII, which we used to calibrate the model parameters through a simulation-optimization process. Our framework also includes a novel variant of VIC that simulates both storage dynamics and operations of water reservoirs. Using the Lancang river basin as a case study, we calibrated

two implementations of VIC, with and without reservoirs. Inline with previous studies (e.g., de Paiva et al. (2013); Abbaspour et al. (2015)), we found that the model without reservoirs attains a reasonable modelling accuracy. In fact, we found that the calibration process of both model implementations yields de facto the same values of the goodness of fit statistics (NSE and TRMSE), suggesting that the model parameterization helps compensates for a structural error, namely the absence of the water reservoirs. More specifically, this effect is achieved by determining the values of six soil parameters ($D_s$, $D_{max}$, $W_s$, b, $d_1$ and

$d_2$) that let this model implementation emulate the presence of water reservoirs.

The first implication of a flawed parameter estimation stands in a poor representation of key hydrological processes, such as surface runoff, infiltration, and baseflow. In our case, we found that, during the dry months, the models calibrated without water reservoirs generate a higher amount of baseflow and runoff than the models with reservoirs. This is an artefact needed to reproduce the higher discharges of hydropower dams that sustain the production of hydro-electricity in the dry season. Vice

versa, baseflow and runoff are reduced during the wet months, so as to account for the decrease in peak flows caused by the fact that dams store part of the water for the following dry season. A poor parameter estimation is also likely to affect several downstream applications of a hydrological model. In our second experiment we exemplify this concept through a climate change impact assessment, in which we contrasted the annual and monthly discharges projected by two selected models (with





and without reservoirs). Both models show a similar trend in the flow regime—i.e., increased monthly discharges during the monsoon season, caused by the projected increase in precipitation—a results found in previous studies (Lauri et al., 2012; Hoang et al., 2016, 2019). Yet, one cannot neglect the different nuances of the flow regime alterations predicted by the two models. In particular, the model with reservoirs presents higher discharges at the peak of the monsoon season than the model

without reservoirs. These nuances may impact some of the conclusions of a climate change impact assessment as well as other model-based studies depending on a reliable estimation of the flow regime.

Naturally, the framework adopted in this study has a few limitations. First, our model calibration focuses solely on six main parameters controlling the rainfall-runoff process, and assumes that the latter are homogeneously distributed across the basin. As explained in Section 3.1.1, the choice of these six parameters is rather established in the literature (Dan et al., 2012; Park

and Markus, 2014; Xue et al., 2015); yet, it is reasonable to assume that the use of more parameters could further improve the model accuracy. As for the use of homogeneously-distributed parameters, our modelling choice is justified by the fact that the use of heterogeneously-distributed parameters would largely impact the computational requirements of the calibration process. We also note that there are no reasons to believe that the use of more (or spatially-distributed parameters) would deeply alter the main findings of this work. Second, our study relies on rule curves conceived to maximize the hydropower production. This is

not a limitation of our study (reservoirs in the Lancang are indeed operated for hydropower supply; Räsänen et al. (2017)), but it certainly constraints the applicability of our modelling framework to regions in which the reservoir operating rules are either available or well understood. It is also a further testimony of the importance of studies aimed to deduct reservoir operating rules from discharge and remote sensed data (Bonnema and Hossain, 2017; Coerver et al., 2018). Third, we focussed our attention on water reservoirs, which are indeed the infrastructures affecting the flow regime in the Lancang. In the lower Mekong basin

(not considered in our spatial domain), the flow regime has been modified not only by the hydropower reservoirs, but also by withdrawals for irrigation supply (Hoang et al., 2019). Looking forward, it would thus be interesting to extend the spatial domain of our model and study how these withdrawals could affect its parameterization.

Overall, the findings of this study reinforce the message that water infrastructures—and their operational settings—play a key role on the reliability of a hydrological modelling exercise, like the quality of the hydro-meteorological data, the model

structure, or the calibration process (Francés et al., 2007; Madsen, 2000). These findings gain further prominence if one considers the expected increase in hydropower development in several regions of the world (Zarfl et al., 2015).

*Author contributions.* TDD and SG conceptualized the paper and its scope. Data collection, model implementation, and experiments were carried out by TDD, with inputs from AFMKC and SG. All authors contributed to the manuscript preparation.

*Data availability.* Precipitation and air temperature data were retrieved from APHRODITE and CFSR datasets, available at http://www.

chikyu.ac.jp/precip/english/ and https://climatedataguide.ucar.edu/climate-data/climate-forecast-system-reanalysis-cfsr. Land use and land cover data were obtained from the GLCC dataset (https://www.usgs.gov/centers/eros/science/), while the soil data were extracted from the



HWSD database (http://www.fao.org/soils-portal/soil-survey/soil-maps-and-databases/harmonized-world-soil-database-v12/en/). The Terra MODIS satellite images (used to calculate the monthly Leaf Area Index and albedo) are available at https://modis.gsfc.nasa.gov. The Landsat TM and ETM+ imagery are available at https://earth.esa.int/web/sppa/mission-performance/esa-3rd-party-missions/landsat-1-7/tm-etm/sensor-description. The global Digital Elevation Model (GTOPO30) is available at http://www.temis.nl/data/gtopo30.html. The GCMs projections were retrieved from https://esgf-node.llnl.gov/projects/esgf-llnl/. All these data are publicly available. The daily discharge data at Chiang Saen and the design specifications of all dams were obtained by the authors from the Mekong River Commission and the International Commission On Large Dams, so they cannot be shared without their consent. Additional data about the dams were retrieved from the Global Reservoir and Dam Database, available at http://globaldamwatch.org/grand/.

*Competing interests.* The authors declare that they do not have individual or collective conflicts of interests.

*Acknowledgements.* This research is supported by Singapore's Ministry of Education (MoE) through the Tier 2 project 'Linking water availability to hydropower supply—an engineering systems approach' (Award No. MOE2017-T2-1-143).





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



**Figure 1.** Mekong river basin (a); and elevation map and location of the hydropower dams in the upper Mekong basin (b). The red squares denote the dams built before 2005 (and therefore included in our study), while the yellow circles indicate the dams built after 2005.





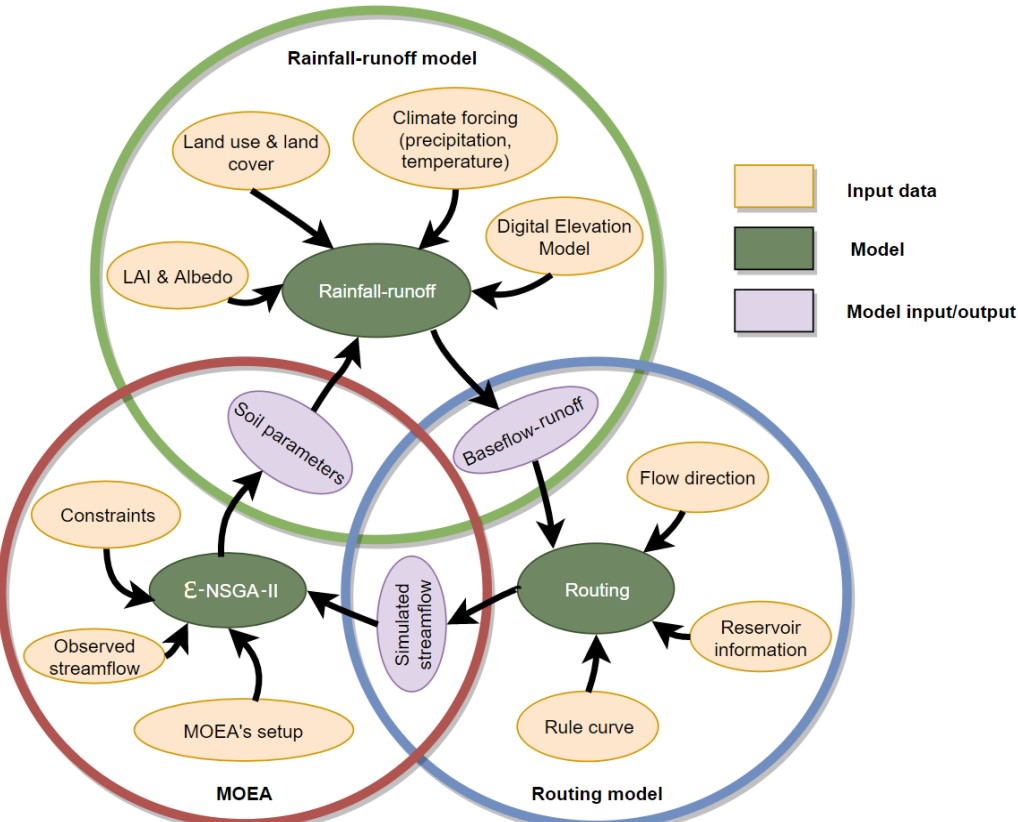

**Figure 2.** Computational framework adopted in the first part of this study. The framework consists of VIC's rainfall-runoff and routing modules and the MOEA $\varepsilon-$NSGAII. The output of the rainfall-runoff module (i.e., gridded baseflow and runoff) is used by the routing module, which simulates the streamflow at multiple locations within the upper Mekong basin. The simulated streamflow is then used to calculate goodness of fit statistics, whose value is optimized with $\varepsilon-$NSGAII by calibrating the parameters of the rainfall-runoff module. In other words, these parameters and goodness of fit statistics represent the decision variables and objective functions used by $\varepsilon-$NSGAII.

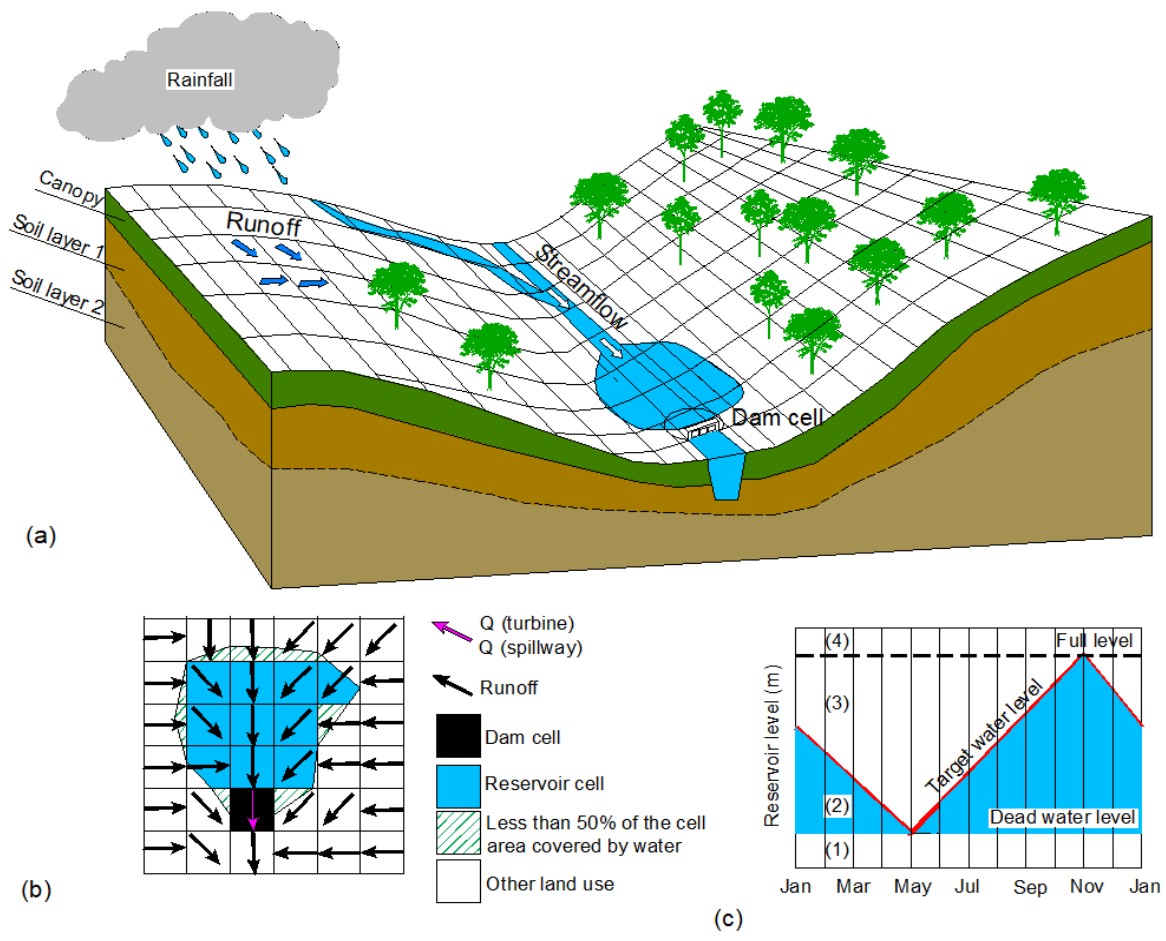

**Figure 3.** Graphical representation of VIC's spatial domain (adapted from http://www.hydro.washington.edu) (a), including the selection of the dam cell (black), reservoir cells (blue), and cells with other land use (white with green lines). The black and pink arrows indicate the direction of the flow routing and discharge from the reservoir (b). Seasonal rule curve (c).

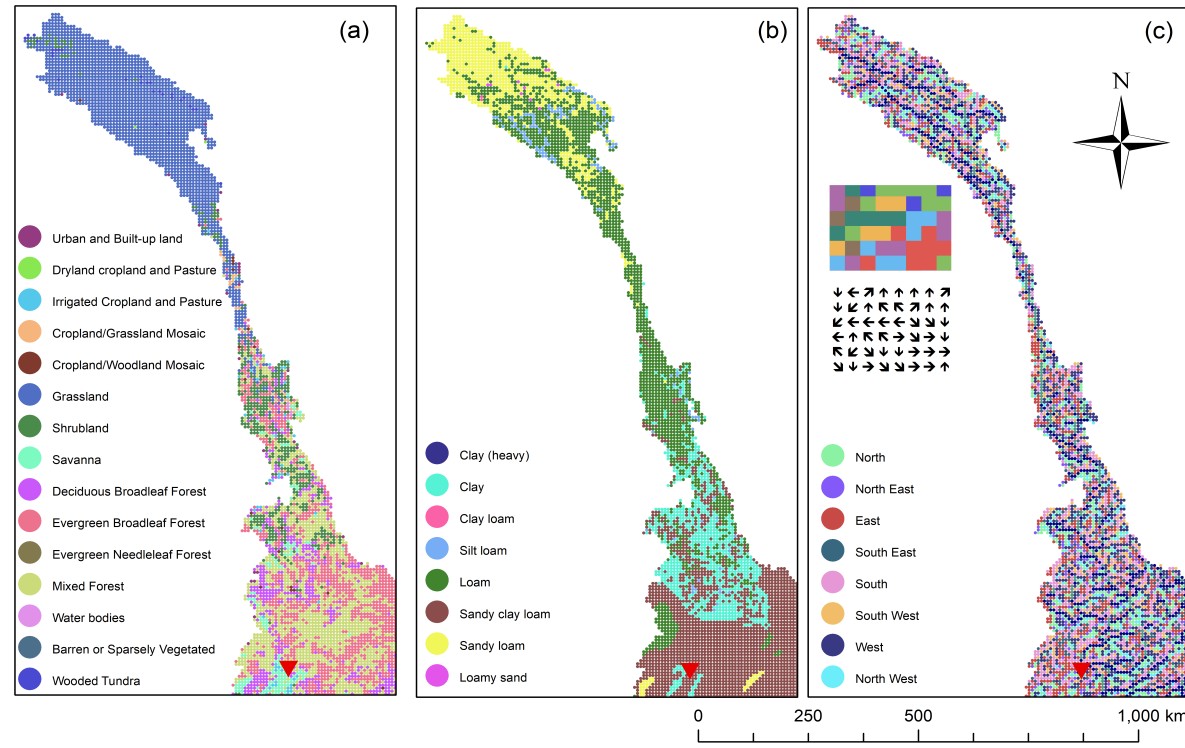

**Figure 4.** Land use map derived from the Global Land Cover Characterization dataset (a); soil map (for the top layer) retrieved from the Harmonized World Soil Database (b); flow direction map (c). The red triangle denotes the position of the Chiang Saen gauging station.





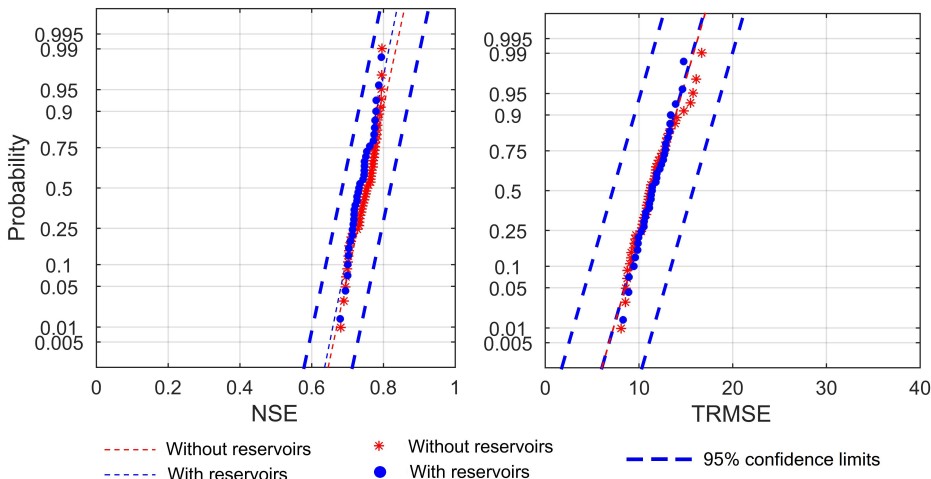

**Figure 5.** Probability plots for the NSE (left) and TRMSE (right) obtained in the model calibration process. The blue circles and red stars specify the results obtained by the models with and without reservoirs, respectively. The dashed blue and red lines represent the theoretical distributions. In both plots, we also report the 95% confidence limits for the models calibrated with reservoirs.



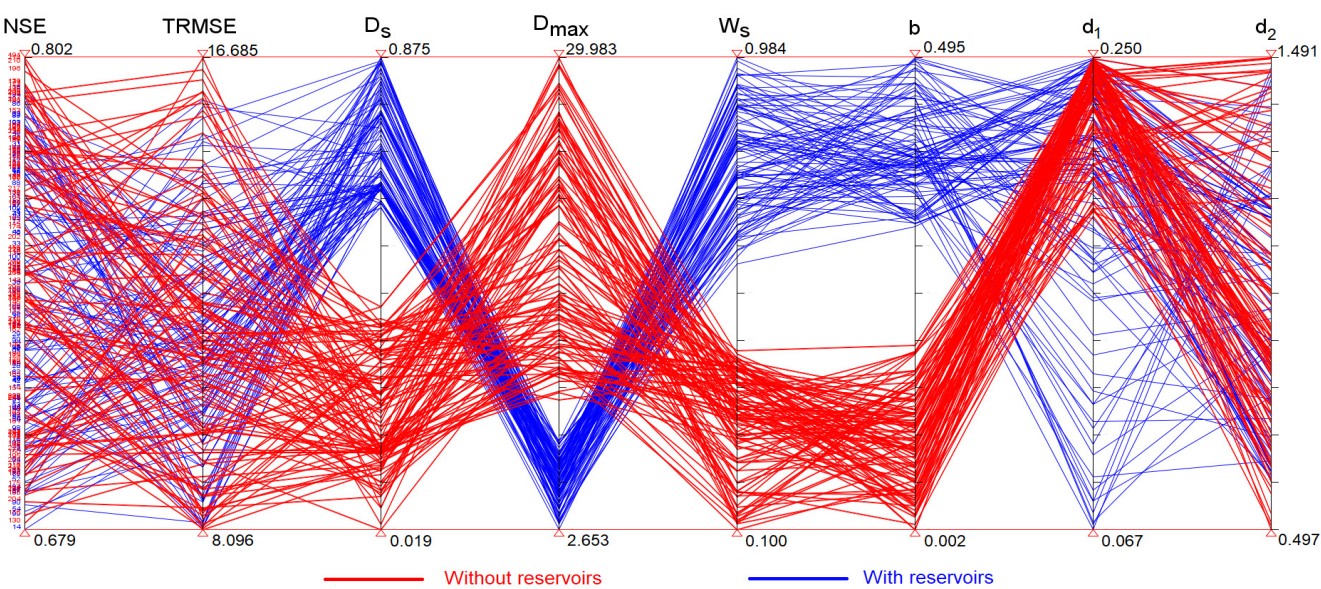

**Figure 6.** Parallel coordinate plot illustrating the values of the goodness of fit statistics (NSE and TRMSE) and model parameters ($D_s$, $D_{max}$, $W_s$, b, $d_1$ and $d_2$) obtained through the optimization-based parameterization exercise. Each line connecting the axes represents a parameterization, along with the corresponding model performance. Blue and red lines denote parameterizations obtained with and without reservoirs.



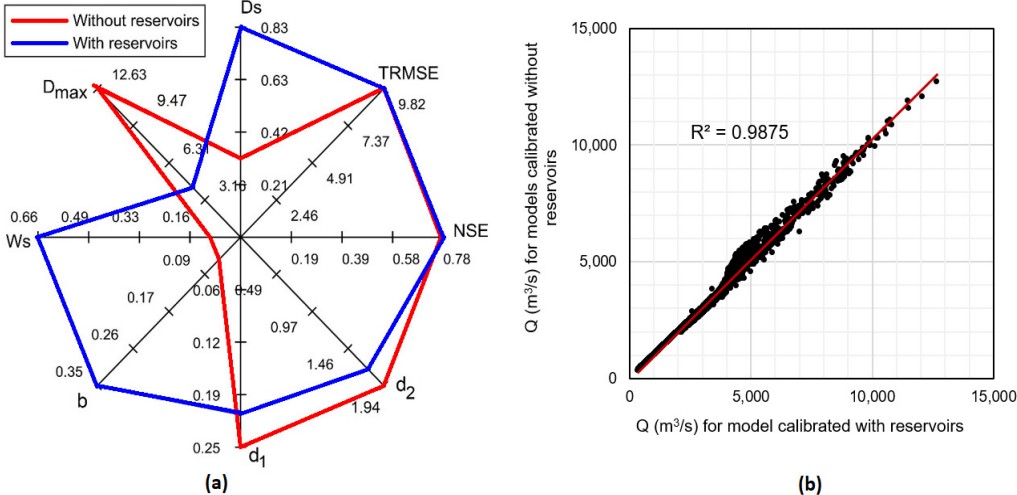

**Figure 7.** Radar chart illustrating the values of Nash-Sutcliffe Efficiency (NSE), Transformed Root Mean Square Error (TRMSE), and model parameters ($D_s$, $D_{max}$, $W_s$, b, $d_1$ and $d_2$) of the two selected models (a); scatter plot comparing the daily discharges simulated by the two models over the periods 1996–2005 (b).





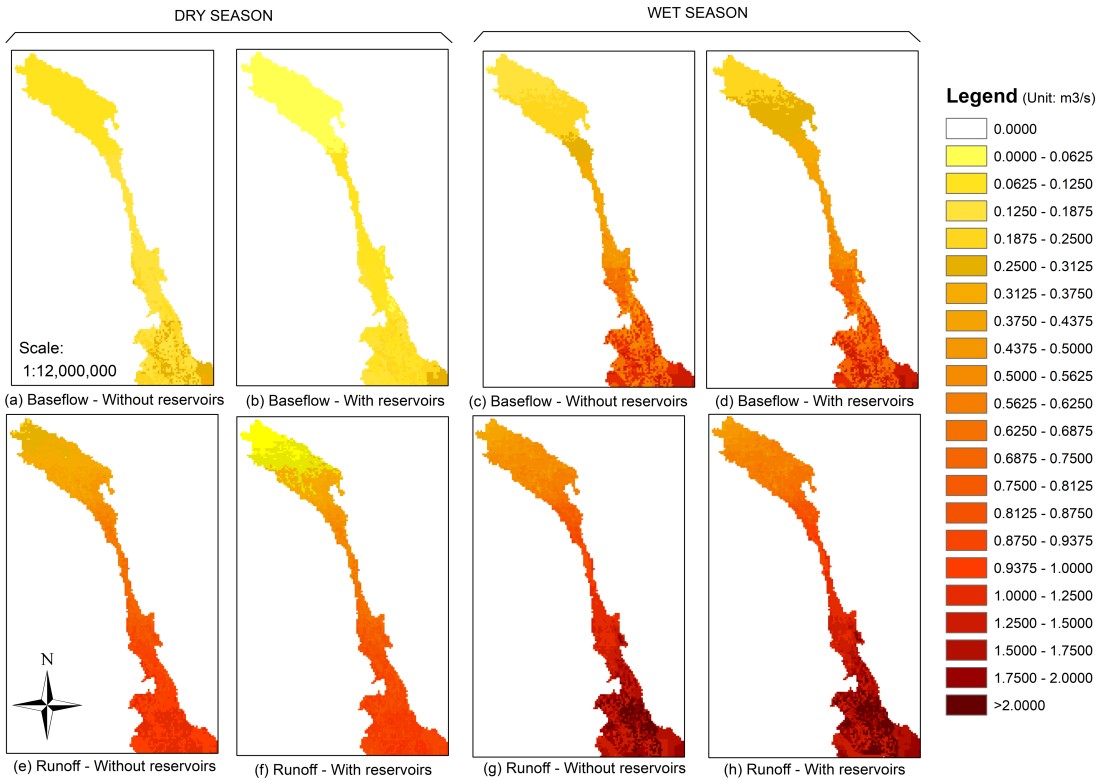

**Figure 8.** Average values of simulated baseflow (top panels) and runoff (bottom panels) simulated by the selected models (with and without reservoirs) during the dry (December–April) and wet (May–November) seasons of the period 1996–2005.



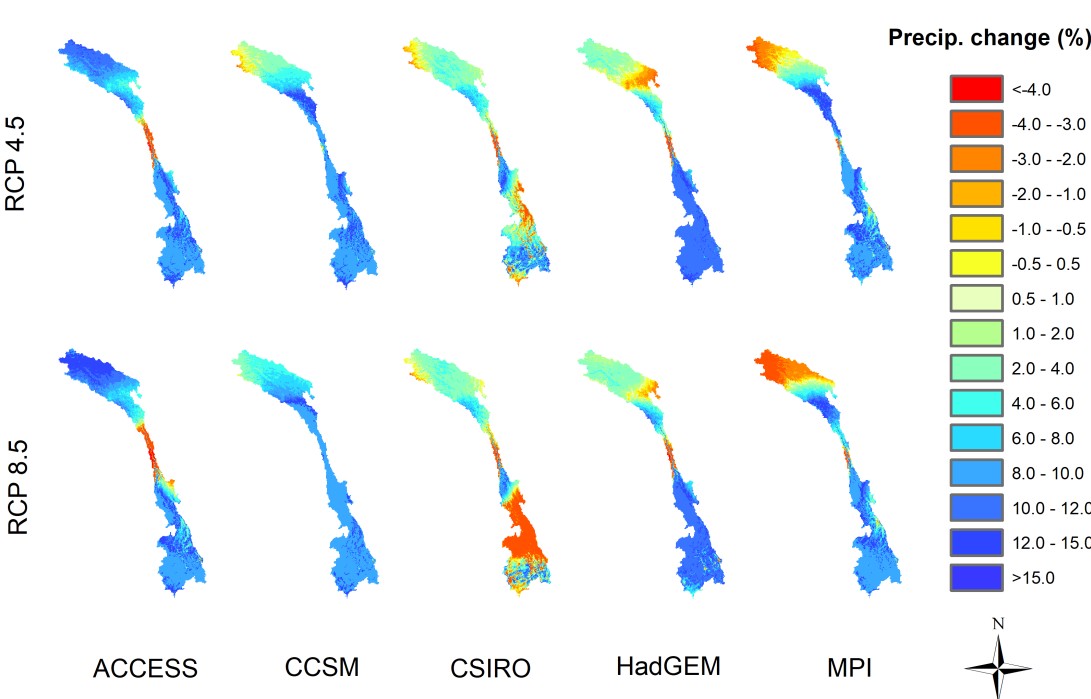

**Figure 9.** Projected changes in total annual precipitation (%) under future climate (2050–2060) compared to the baseline (1996–2005).





**Figure 10.** Projected changes in daily maximum and minimum temperature under future climate (2050–2060) compared to the baseline (1996–2005). These changes are produced by five Global Circulation Models (GCMs) and two Representative Concentration Pathways (RCPs).



**Figure 11.** Projected monthly discharges at Chiang Saen under five GCMs and two RCPs for the two selected models calibrated without and with reservoirs (a-d). Box plots highlighting the variability in monthly discharges predicted by the two models under RCP 4.5 (e) and RCP 8.5 (f).



**Table 1.** Main parameters controlling the rainfall-runoff process in VIC. The third column contains the range of each parameter value considered during the calibration process.

| Name | Unit | Feasible range | Description |
|------|------|----------------|-------------|
| $d_1$ | m | [0.05, 0.25] | Thickness of the upper soil layer |
| $d_2$ | m | [0.3, 1.5] | Thickness of the lower soil layer |
| b | - | (0, 0.9] | Variable Infiltration Capacity curve parameter |
| $D_{max}$ | mm/day | (0, 30] | Maximum baseflow |
| $D_s$ | - | (0, 1) | Fraction of $D_{max}$ where non-linear baseflow begins |
| $W_s$ | - | (0, 1) | Fraction of maximum soil moisture where non-linear baseflow occurs |





**Table 2.** Design specifications of the dams implemented in our VIC model (simulation period 1995–2005). The term Year denotes the time at which each reservoir became operational.

| No. | Name | Year | Long. (°E) | Lat. (°N) | Height (m) | Storage (Mm$^3$) | Design discharge (m$^3$/s) | Inst. cap. (MW) |
|---|---|---|---|---|---|---|---|---|
| 1 | Xi'er He 4 | 1971 | 100.066 | 20.000 | 20 | 14 | 283 | 50 |
| 2 | Xi'er He 1 | 1989 | 100.202 | 30.000 | 30 | 1,501 | 60 | 105 |
| 3 | Xi'er He 2 | 1987 | 100.131 | 25.562 | 37.25 | 0.2 | 168 | 50 |
| 4 | Xi'er He 3 | 1988 | 100.108 | 20.700 | 20.70 | 0.09 | 304 | 50 |
| 5 | Manwan | 1992 | 100.446 | 24.625 | 136 | 257 | 1,700 | 1,670 |
| 6 | Longdi | 1997 | 99.724 | 26.221 | 95 | 13.30 | 12.34 | 10 |
| 7 | Laoyinyan | 1997 | 99.818 | 24.469 | 4.31 | 10.92 | 9.3 | 16 |
| 8 | XunCun | 1999 | 99.993 | 25.422 | 67 | 73.74 | 146 | 78 |
| 9 | Jinfeng | 1998 | 101.225 | 21.592 | 45 | 19.48 | 45 | 16 |
| 10 | Dachaoshan | 2003 | 100.370 | 24.025 | 115 | 367 | 2,109 | 1,350 |
| 11 | Jinhe | 2004 | 97.333 | 34.000 | 34 | 4.27 | 222 | 60 |





**Table 3.** CMIP5 GCMs used for the climate change impact assessment.

| GCM | Spatial resolution (long × lat) | Control baseline | Developer |
|---|---|---|---|
| ACCESS1-0 | $1.875° \times 1.25°$ | 1850–2006 | Commonwealth Scientific and Industrial Research Organization, Australia |
| CCSM4 | $1.25° \times 0.94^0$ | 1850–2005 | National Center for Atmospheric Research, USA |
| CSIRO Mk3.6 | $1.875° \times 1.875°$ | 1850–2005 | Commonwealth Scientific and Industrial Research Organization and the Queensland Climate Change Centre of Excellence, Australia |
| HadGEM2 ES | $1.875° \times 1.24°$ | 1861–2010 | Met Office, UK |
| MPI-ESM-LR | $1.875° \times 1.875°$ | 1850–2005 | Max Planck Institute for Meteorology, Germany |





**Table 4.** Relative changes in annual river discharges at the Chiang Saen station for the future period (2050–2060) relative to the reference one (1996–2005). The lowest and highest changes are presented with the corresponding scenarios. The results reported in the first and second rows were produced by the selected models without and with reservoirs, respectively.

| Scenario | RCP 4.5 | | RCP 8.5 | |
|---|---|---|---|---|
| | Ensemble mean (%) | Range (%) | Ensemble mean (%) | Range (%) |
| Without reservoirs | +13.62 | +6.36 to +23.66 CSIRO–ACCESS | +13.92 | -0.67 to +28.89 CSIRO–ACCESS |
| With reservoirs | +13.56 | +6.28 to +23.56 CSIRO–ACCESS | +13.83 | -0.63 to +28.68 CSIRO–ACCESS |