# Peer review of "On the representation of water reservoir storage and operations in large-scale hydrological models: implications on model parameterization and climate change impact assessments"

_Hydrology and Earth System Sciences, 2019_

## Referee Comment (RC1) · Anonymous Referee #1 · 14 Aug 2019

This paper presents a computational framework based on the Variable Infiltration Capacity (VIC) model and a Multi-Objective Evolutionary Algorithm that enables to analyze the effects of water reservoir representation on the parameterization of hydrological models. The modelling approach was applied to the upper Mekong river basin, upstream the Chiang Saen gauging station, considering two configuration schemes with and without water reservoirs. The authors exposed the theme in a clear, logical sequence, which resulted in a well-written comprehensive text. In my opinion, this work should be accepted for publication at the Hydrology and Earth System Sciences

(HESS) journal after minor revisions. Attached you find some comments/suggestions. Introduction: A comprehensive literature review was done concerning hydrological modelling applied to large river basins, highlighting the limited number of model approaches that enables the direct representation of reservoir water storage with target operating rules to achieve pre-defined objectives. Furthermore, the scientific contribution of this manuscript is clearly defined from line 21 to 34 (Page 03). Study area: The study area description looks poor. Please include more information concerning soil properties and classes, land use, geology and geomorphology, just to enable a better comprehension of main driving forces related to hydrological processes in the Upper Mekong river basin. In Section 3.2.1 you presents the way the input variables (Land use and land cover data, soil data, and flow direction are obtained/processed, but a discussion about those data is lacking. Materials and methods: The way used to achieve the objectives of the manuscript is clearly presented and well-organized. Nevertheless. I would suggest including a sensitivity analysis of the model results to the parameters controlling the rainfall-runoff process (Ds, Dmax, Ws, b, d1 and d2) in the Variable Infiltration Capacity model. Results: The questions raised in the introduction section were answered. The results showed that a flawed model parametrization by disregarding anthropogenic interventions (such as hydraulic infrastructure) led to the overestimation of baseflow and runoff during the dry season to compensate water release related to hydropower production. Despite this, I would suggest discussing more deeply the model parameterization and results. According to the authors, the VIC model has been previously used by other researchers. Could your results be compared with them? Or other models with similar purpose?

Specific comments are listed as follows: 1) Page 04, line 07: How large is the Upper Mekong river basin? Please include the catchment area in km$^2$. 2) Page 04, lines 07-08: The information of elevation ranging from 362 m to 6,494 m should be included. 3) Page 04, line 30: Please include a complementary information explaining why the hydrological alterations became more evident since 1992. The largest reservoir (Xi'er He 1) was built before in 1989. 4) Page 05, line 19 and Table 1: How the feasible ranges

of the two çãosoil layers thickness (d1 and d2) were defined? 5) Page 06, line 29: How the target water level is defined for each reservoir? 6) Page 08 lines 02-17: What processes are considered in the reservoir water balance and operation? Are infiltration loss and groundwater inflow disregarded? 7) Page 09, line 11: "parellelized" should be changed to "parallelized". 8) Page 11, lines 05-24: Please discuss the pattern observed in Figure 6, with different ranges of the model parameters for the simulation with and without reservoirs. Some parameters presented a more spread pattern for the scenario with reservoirs and a more uniform one for the scenario without reservoir and vice versa. 9) Fig. 2: Please use the term "modelling approach" instead "model" (Figure legend). 10) Fig. 3: Please include a legend describing the four levels highlighted in Figure 3c. 11) Fig 8: I would suggest to reduce the number of baseflow intervals to enable a better visualization of model results related to the configuration scheme with and without reservoirs.

---

## Referee Comment (RC2) · Anonymous Referee #2 · 2 Sep 2019

General comments:

This work contributes to literature on the representation of human interactions in hydrological model by demonstrating the importance of modeling reservoirs. Using the high-profile case of the Lancang / Mekong river basin, they calibrate a large-scale hydrological model (VIC), both with and without reservoirs. They show that while there exist parameter sets for which the model can be calibrated in both cases, the without-reservoir model compensate the absence of human-operated storage by artificially creating soil storage. Then they go on to investigate the reactions of both models

to climate change.

This is an interesting, timely and well-written piece of work that fits well within the scope of HESS. My comments are mainly at how the results are handled and interpreted:

1) While authors explain very well the consequences of the no-reservoir calibration in terms of structural model behavior, they do not show what that means for water resources appraisals. Indeed, both models are calibrated to have the same behavior downstream of 1 gauge. The crucial difference is that the model including reservoirs can be 1) validated at other (upstream) sites, and 2) validated with post 2005 dams (or the results can be extrapolated with new dams or new operating rules). This point is not really demonstrated.

2) Likewise, I disagree that the results from climate models are very different. Yes, there are some small difference, and authors do a good job of explaining them, but these differences are arguably small compared with the uncertainty surrounding downscaled climate projections. In fact, it is remarkable how robust the no-reservoir model results are with climate change.

3) I would advise authors to try and show how both calibrations differ at upstream gauges (where only the with-reservoir calibration would yield sensible results) where there is data, then carry on that comparison with climate change.

4) Alternatively (but this is probably more work), authors can focus on post-2005 years with the addition of new dams and look at the differences between both model at the outlet. And again look at how the two models differ with climate change then

5) Figure 8 is not very clear and could be replaced with curves comparing average monthly basinwide runoff for the two calibrations (and same for baseflow). This would show explicitly how the no-reservoir calibration compensates for the reservoirs. Alternatively, authors could map the dry / wet season cell-by-cell differences in baseflow / runoff between the two calibrations.
6) Figures 9 and 10 don't bring much and could be relegated to supplementary material to make place for new figures that can show the consequences of omitting reservoirs, e.g., as suggested in 3) and 4)

Detailed comments:

Figure 1.b: the scale, useful in 1.a, is missing.

Section 3.1.2: a short description of the version 4.2 of VIC's routing module would be helpful here. Recall that this research is useful for all large-scale hydrological models, not just for VIC experts.

Page 8, lines 2-9: there is a recent publication presenting a database of reservoir storage-area-depth relationships in Yigzaw et al (2018) there https://agupubs.onlinelibrary.wiley.com/doi/10.1029/2017WR022040 could that affect your results in significant ways?

Page 9, lines 8-12: how many parallel processors are running? (l. 12) by runtime, do you mean the wall clock time or the total computational time used by all the processors (i.e. wallclock time times # of processors)? How do we know that 20 seeds are enough? And that the algorithm has converged after 250 function evaluations? Finally, what is the runtime for one run of VIC for that basin?

Page 14, lines 14-17: I disagree this is a real limitation, since proposing a universal rule system for reservoir operations is not an aim of this paper. Instead, a consequence of your work is that hydrological model calibration with reservoirs and a bespoke release rule means that the model still captures key hydrological processes once the release rules change (because there are new reservoirs / because reservoirs' purposes evolve). In contrast, if reservoirs are not represented to begin with, once the number and / or operations of reservoirs evolves, the model's hydrological parameters have to be recalibrated every time.
* * *
334, 2019.
Interactive
comment

---

## Author Comment (AC1) · 25 Sep 2019

General Comments

R: This paper presents a computational framework based on the Variable Infiltration Capacity (VIC) model and a Multi-Objective Evolutionary Algorithm that enables to analyze the effects of water reservoir representation on the parameterization of hydrological models. The modelling approach was applied to the upper Mekong river basin, upstream the Chiang Saen gauging station, considering two configuration schemes

with and without water reservoirs. The authors exposed the theme in a clear, logical sequence, which resulted in a well-written comprehensive text. In my opinion, this work should be accepted for publication at the Hydrology and Earth System Sciences (HESS) journal after minor revisions. Attached you find some comments/suggestions.

A: We thank the reviewer for the positive comments and practical suggestions for improving the manuscript.

R: Introduction: A comprehensive literature review was done concerning hydrological modelling applied to large river basins, highlighting the limited number of model approaches that enables the direct representation of reservoir water storage with target operating rules to achieve pre-defined objectives. Furthermore, the scientific contribution of this manuscript is clearly defined from line 21 to 34 (Page 03).

A: Thanks for the positive feedback.

R: Study area: The study area description looks poor. Please include more information concerning soil properties and classes, land use, geology and geomorphology, just to enable a better comprehension of main driving forces related to hydrological processes in the Upper Mekong river basin.

A: Yes, we will include more details regarding the soil properties, land use, geology and geo-morphology. Specifically, we will include the following paragraph in Section 2: "The aforementioned orography and climate conditions are not particularly suitable for agricultural activities, which are indeed limited. The basin is mountainous, with mostly rocks and a shallow Quaternary alluvium (Gupta, 2009; Carling, 2009). Due to the impermeability of bedrock underneath isolated valleys, only a very small fraction of water leaks into the ground through karst aquifer units (Lee et al., 2017). As a result, subsurface water is mostly generated in the shallow loam layer in the form of baseflow. Reservoirs are created by roller-compacted concrete gravity dams in narrow river sections with rocky abutments. Thus, foundation, abutments, and dams can be considered impervious."

[Figure]

R: In Section 3.2.1 you present the way the input variables (Land use and land cover data, soil data, and flow direction) are obtained/processed, but a discussion about those data is lacking.

A: We will add the following explanations in Section 3.2.1:

- Second paragraph (on land use and land cover data): "In the basin, broadleaf forest and grassland dominate other types of land cover. The upper part of the catchment has high elevations, so vegetation is mostly grassland. The lower reaches have complex terrain and large altitudinal variations, resulting in mixed coniferous forest ecoregions."
- Second paragraph (on soil data): "In the Chinese part of the basin, soil is characterized by a shallow layer consisting of loam, sandy loam, and clay. At the border between China, Myanmar, and Laos (near Chiang Saen station), soil characteristics are dominated by the presence of sandy clay loam." - Third paragraph (on flow direction data): "Following the main direction of the Mekong river and altitudinal gradient of the region, the flow direction is predominantly southward."

R: Materials and methods: The way used to achieve the objectives of the manuscript is clearly presented and well-organized. Nevertheless, I would suggest including a sensitivity analysis of the model results to the parameters controlling the rainfall-runoff process (Ds, Dmax, Ws, b, d1 and d2) in the Variable Infiltration Capacity model.

A: Thanks for the suggestion. We will include an analysis to explore the sensitivity of the model output with respect to the values of these six parameters (for both model configurations). Specifically, we will proceed by linking VIC (with and without reservoirs) with SAFE toolbox (Pianosi et al., 2015), which provides a number of global sensitivity algorithms. In order to limit the number of figures and tables in the main manuscript, we will likely include the sensitivity analysis results in the Supplement.

R: Results: The questions raised in the introduction section were answered. The results showed that a flawed model parametrization by disregarding anthropogenic interventions (such as hydraulic infrastructure) led to the overestimation of baseflow and

runoff during the dry season to compensate water release related to hydropower production. Despite this, I would suggest discussing more deeply the model parameterization and results. According to the authors, the VIC model has been previously used by other researchers. Could your results be compared with them? Or other models with similar purpose.

A: Literature offers a few studies that implemented VIC model to the Mekong River basin (e.g., Zhong et al. (2019), Chang et al. (2019)). Unfortunately, these studies do not report details about the parameterization–they tend to focus on the modelling accuracy. The same comment applies to studies that applied other hydrological models to the Mekong (e.g., Lauri et al. (2012), Rasanen et al. (2012), Hoang et al. (2016)). For this reason, we cannot carry out a direct comparison between our parameterization and the one obtained by previous studies.

Specific Comments:

R: 1) Page 04, line 07: How large is the Upper Mekong river basin? Please include the catchment area in km2.

A: The Upper Mekong river basin has an area of about 167,400 km2; that's about 24% of the Mekong's basin area. We will add this information to the revised manuscript.

R: 2) Page 04, lines 07-08: The information of elevation ranging from 362 m to 6,494 m should be included.

A: We will add this information.

R: 3) Page 04, line 30: Please include a complementary information explaining why the hydrological alterations became more evident since 1992. The largest reservoir (Xi'er He 1) was built before in 1989.

A: Previous works focusing on the hydrological alterations caused by hydropower dams in the Lancang basin generally consider two periods, namely pre- and post 1991 (e.g., 1960-1991 and 1992-2013) (Cochrane et al. (2014), Lu et al. (2014), Dang et al.

(2016)). The reason for this choice is due to Manwan dam, which received significant public attention because of its location. It is indeed the first dam built on the main stem of the river. We understand that our sentence is potentially misleading, so we will clarify this point in the revised version of the manuscript.

R: 4) Page 05, line 19 and Table 1: How the feasible ranges of the two soil layers thickness (d1 and d2) were defined?

A: The definition of the feasible ranges is based on the ranges that were adopted in other studies applying VIC model to large catchments (e.g., Dan et al. (2012), Park and Markus (2014), Xue et al. (2015)). Note that the same ranges are also recommended by Wi et al. (2017), who developed a general-purpose user-friendly software package for VIC. We will clarify this point in the caption of Table 1.

R: 5) Page 06, line 29: How the target water level is defined for each reservoir?

A: As explained at Page 6 (Line 12-21), the target water level is defined as follows. First, we determine the minimum and maximum water levels that a reservoir should reach within a year. In our case, we use the minimum and maximum elevation levels, which are given in the design specifications of each reservoir. Then, we set the time at which the minimum and maximum water levels should be reached. In our case, we use the months of May and November, which correspond to the beginning and end of the wet season. Finally, we connect these points with a piecewise linear function that gives us the daily target level for each calendar day. With these rule curves, we allow the water level to drawdown during the pre-monsoon months and to recharge during the wet months, thereby maximizing the hydropower production–for further details, please refer to Piman et al. (2012).

R: 6) Page 08 lines 02-17: What processes are considered in the reservoir water balance and operation? Are infiltration loss and groundwater inflow disregarded?

A: We considered three main processes, namely inflow, release, and evaporation. As

noted by the reviewer, there are two other processes that could be considered, that is, infiltration and seepage (via dam body, abutment, and foundation). The reason for which we disregarded them is twofold. First, the Upper Mekong basin is a mountainous region, with mostly rocks and a shallow Quaternary alluvium (Gupta, 2009; Carling, 2009), so the infiltration losses are to some extent marginal as compared to inflow, release, and evaporation. Second, the dams considered in our study are built with concrete (and with rocky abutments and foundations), so seepage is indeed limited. We will touch upon this point at the beginning of Section 3.1.2 and refer to the expanded discussion on soil properties, land use, geology and geo-morphology (please refer to our reply to the first comment).

R: 7) Page 09, line 11: "parellelized" should be changed to "parallelized".

A: Thanks for spotting this typo, which we will correct in the revised manuscript.

R: 8) Page 11, lines 05-24: Please discuss the pattern observed in Figure 6, with different ranges of the model parameters for the simulation with and without reservoirs. Some parameters presented a more spread pattern for the scenario with reservoirs and a more uniform one for the scenario without reservoir and vice versa.

A: Thanks for raising this point. As mentioned in our reply to the third comment, we will carry out a global sensitivity analysis for both model setups (i.e., with and without reservoirs). We believe such analysis will help us expand the discussion on Figure 6.

R: 9) Fig. 2: Please use the term "modelling approach" instead "model" (Figure legend).

A: We think that the term "model" is more appropriate, since the picture depicts two specific numerical models (i.e., VIC's rainfall-runoff and routing modules) and an optimization algorithm (i.e., epsilon-NSGA-II)–and not a family of similar models that could be denoted with the term "approach".

R: 10) Fig. 3: Please include a legend describing the four levels highlighted in Figure

3c.

A: Thanks for the suggestion. Yes, we will include either a legend or a detailed explanation in the caption.

R: 11) Fig 8: I would suggest to reduce the number of baseflow intervals to enable a better visualization of model results related to the configuration scheme with and without reservoirs.

A: Thanks for the suggestion. We will modify the number of baseflow and runoff intervals.

References

Carling, P. A. (2009). The geology of the lower Mekong River. The Mekong. Academic Press, 13-28.

Chang, C.H., Lee, H., Hossain, F., Basnayake, S., Jayasinghe, S., Chishtie, F., Saah, D., Yu, H., Sothea, K., Du Bui, D. (2019). A model-aided satellite-altimetry-based flood forecasting system for the Mekong River. Environmental Modelling & Software, 1:112-27.

Cochrane, T., Arias, M., Piman, T. (2014). Historical impact of water infrastructure on water levels of the Mekong River and the Tonle Sap system, Hydrology and Earth System Sciences, 18, 4529–4541.

Dan, L., Ji, J., Xie, Z., Chen, F., Wen, G., Richey, J. E. (2012). Hydrological projections of climate change scenarios over the 3H region of China: A VIC model assessment, Journal of Geophysical Research: Atmospheres, 117.

Dang, T. D., Cochrane, T. A., Arias, M. E., Van, P. D. T., de Vries, T. T. (2016). Hydrological alterations from water infrastructure development in the Mekong floodplains, Hydrological Processes, 30, 3824–3838.

Gupta, A. (2009). Geology and landforms of the Mekong Basin. The Mekong. Academic Press, 29-51.

Hoang, L.P., Lauri, H., Kummu, M., Koponen, J., van Vliet M., Supit, I., Leemans, R., Kabat, P., Ludwig, F. (2016). Mekong River flow and hydrological extremes under climate change. Hydrology and Earth System Sciences, 20:3027-41.

Lauri, H., Moel, H. D., Ward, P. J., Räsänen, T. A., Keskinen, M., Kummu, M. S. (2012). Future changes in Mekong River hydrology: impact of climate change and reservoir operation on discharge. Hydrology and Earth System Sciences, 16: 4603-4619.

Lee, E., Ha, K., Ngoc, N. T. M., Surinkum, A., Jayakumar, R., Kim, Y., Hassan, K. B. (2017). Groundwater status and associated issues in the Mekong-Lancang River Basin: international collaborations to achieve sustainable groundwater resources. Journal of Groundwater Science and Engineering, 5(1), 1-13.

Lu, X., Li, S., Kummu, M., Padawangi, R., Wang, J. 2014. Observed changes in the water flow at Chiang Saen in the lower Mekong: Impacts of Chinese dams?, Quaternary International, 336, 145–157.

Park, D., Markus, M. (2014) Analysis of a changing hydrologic flood regime using the Variable Infiltration Capacity model, Journal of Hydrology, 515, 267–280.

Pianosi, F., Sarrazin, F., Wagener, T. (2015). A Matlab toolbox for global sensitivity analysis. Environmental Modelling & Software, 70, 80-85.

Piman, T., Cochrane, T., Arias, M., Green, A., Dat, N. (2012). Assessment of flow changes from hydropower development and operations in Sekong, Sesan, and Srepok rivers of the Mekong basin. Journal of Water Resources Planning and Management, 139, 723–732.

Räsänen, T. A., Koponen, J., Lauri, H., Kummu, M. (2012). Downstream hydrological impacts of hydropower development in the Upper Mekong Basin. Water Resources Management, 26(12), 3495-3513.

Wi, S., Ray, P., Demaria, E. M., Steinschneider, S., Brown, C. (2017). A user-friendly software package for VIC hydrologic model development. Environmental Modelling & Software, 98, 35-53.

Xue, X., Zhang, K., Hong, Y., Gourley, J. J., Kellogg, W., McPherson, R. A., Wan, Z., Austin, B. N. (2015). New multisite cascading calibration approach for hydrological models: Case study in the red river basin using the VIC model, Journal of Hydrologic Engineering, 21, 05015 019.

Zhong, R., Zhao, T., He, Y., Chen, X. (2019). Hydropower change of the water tower of Asia in 21st century: A case of the Lancang River hydropower base, upper Mekong. Energy, 179, 685-696.
* * *

---

## Author Comment (AC2) · 25 Sep 2019

General Comments:

R: This work contributes to literature on the representation of human interactions in hydrological model by demonstrating the importance of modeling reservoirs. Using the high-profile case of the Lancang / Mekong river basin, they calibrate a large-scale hydrological model (VIC), both with and without reservoirs. They show that while there exist parameter sets for which the model can be calibrated in both cases, the without-

reservoir model compensate the absence of human-operated storage by artificially creating soil storage. Then they go on to investigate the reactions of both models to climate change.

This is an interesting, timely and well-written piece of work that fits well within the scope of HESS. My comments are mainly at how the results are handled and interpreted.

A: We thank the reviewer for the positive feedback.

R: 1) While authors explain very well the consequences of the no-reservoir calibration in terms of structural model behavior, they do not show what that means for water resources appraisals. Indeed, both models are calibrated to have the same behavior downstream of 1 gauge. The crucial difference is that the model including reservoirs can be 1) validated at other (upstream) sites, and 2) validated with post 2005 dams (or the results can be extrapolated with new dams or new operating rules). This point is not really demonstrated.

A: Thanks for this suggestion. We totally agree that we can strengthen our message by implementing one of these two validations. In particular, we believe that the first option is the most feasible, although it presents some minor challenges on which we elaborate below. (Our thoughts regarding the second option are outlined as a response to comment no. 4.)

Discharge data in the Chinese section of the Mekong basin are not available to the international research community. More specifically, only water level data for the flood season (June-October) at Yunjinghong station (located downstream of Jinghong reservoir; see Figure 1 in the manuscript) are available through the Mekong River Commission (http://www.mrcmekong.org/news-and-events/news/mrc-and-china-renew-pact-on-water-data-provision-and-other-cooperation-initiatives/). Therefore, we opted for retrieving data from three previous studies (He et al. (2009), Wang et al. (2018), and Tang et al. (2019)) that published monthly discharge values at Jiuzhou station (located near Gonguoqiao reservoir; see Figure 1) for the period 1996-2005.

[Figure]

When retrieving (digitalizing) the data, we found a maximum discrepancy between the three time series of about 10 m3/s; a negligible value if we consider that the minimum discharge is roughly 400 m3/s.

With this 10-year time series, we can then carry out a thorough validation exercise, whose result is illustrated in Figure 1 (of this document). We can note that the model calibrated without reservoirs largely overestimates the dry season flow and slightly underestimates the wet season flow; a result confirmed by the values of NSE and TRMSE. This result corroborates our finding that during the dry season the model without reservoirs generates more baseflow and runoff than the model with reservoirs (and vice versa during the wet season).

R: 2) Likewise, I disagree that the results from climate models are very different. Yes, there are some small difference, and authors do a good job of explaining them, but these differences are arguably small compared with the uncertainty surrounding down-scaled climate projections. In fact, it is remarkable how robust the no-reservoir model results are with climate change.

A: We agree with the reviewer that the uncertainty surrounding the climate change projections is larger than the difference (in discharge) due to the representation of water reservoirs in VIC. This said, we also note that such difference is non-negligible and consistent across both RCPs. We will thus proceed by slightly toning down our conclusions on this result in the revised version of the manuscript.

R: 3) I would advise authors to try and show how both calibrations differ at upstream gauges (where only the with-reservoir calibration would yield sensible results) where there is data, then carry on that comparison with climate change.

A: Yes, we will include an additional validation based on data at an upstream gauge, as explained in our reply to comment no. 1. We will also include the same comparison under the climate change scenarios.

R: 4) Alternatively (but this is probably more work), authors can focus on post-2005 years with the addition of new dams and look at the differences between both model at the outlet. And again look at how the two models differ with climate change then.

A: The validation based on discharge data collected in the upstream reaches of the basin is indeed simpler to implement, so we prefer to adopt such option. This said, we understand the potential of this second suggested validation, so we will elaborate on this point in the concluding remarks (please also refer to our answer to the last minor comment).

R: 5) Figure 8 is not very clear and could be replaced with curves comparing average monthly basinwide runoff for the two calibrations (and same for baseflow). This would show explicitly how the no-reservoir calibration compensates for the reservoirs. Alternatively, authors could map the dry / wet season cell-by-cell differences in baseflow / runoff between the two calibrations.

A: Yes, we agree that Figure 8 is indeed not clear. The option of replacing it with curves comparing average monthly basin-wide runoff for the two calibrations is sound. Yet, such figure would not represent explicitly the spatial variability or runoff and baseflow; something that we deem important. We thus prefer to keep the current 'skeleton' of Figure 8, which we will improve with two changes: 1) modify the number of baseflow intervals (as suggested by Reviewer #1), and 2) use the average annual values of simulated baseflow and runoff (instead of seasonal). A preliminary version of the new figure is given in Figure 2) (of this document).

R: 6) Figures 9 and 10 don't bring much and could be relegated to supplementary material to make place for new figures that can show the consequences of omitting reservoirs, e.g., as suggested in 3) and 4).

A: Yes, we will move both Figure 9 and 10 to the supplement.

Detailed Comments:

R: Figure 1.b: the scale, useful in 1.a, is missing.

A: We will add the scale to Figure 1.b.

R: Section 3.1.2: a short description of the version 4.2 of VIC's routing module would be helpful here. Recall that this research is useful for all large-scale hydrological models, not just for VIC experts.

A: We will provide a brief description of VIC's routing module. However, we think it's better to include such description in Section 3.1.1, which focuses entirely on VIC (section 3.1.2 is about the modification we introduced). More specifically, we will add the following paragraph: "This module takes as input the gridded surface runoff and baseflow produced by the rainfall-runoff module. For each cell, it then implements a Unit Hydrograph approach to transport the runoff and baseflow to the outlet of that grid–under the assumption that all runoff exits a cell in a single flow direction. Finally, it simulates the channel routing using the linearized Saint-Venant equation."

R: Page 8, lines 2-9: there is a recent publication presenting a database of reservoir storage-area-depth relationships in Yigzaw et al (2018) there https://agupubs.onlinelibrary.wiley.com/doi/10.1029/2017WR022040 could that affect your results in significant ways?

A: Thanks for pointing us to this paper. We noticed that only Manwan reservoir is in this new database, so we believe this cannot impact our model implementation / results. This said, we believe that the adoption of Liebe's method should provide robust results–the method has been tested and adopted by several state-of-the-art studies in large-scale hydrology.

R: Page 9, lines 8-12: how many parallel processors are running? (l. 12) by runtime, do you mean the wall clock time or the total computational time used by all the processors (i.e. wallclock time times # of processors)? How do we know that 20 seeds are enough? And that the algorithm has converged after 250 function evaluations? Finally, what is

the runtime for one run of VIC for that basin?

A: For each (of the 20) seed(s), we used four cores. As for the runtime, we reported the wall-clock time per core. The runtime of VIC for the basin is about 50 mins (on one core). We will clarify these details in the revised version of the manuscript.

There are no specific guidelines in the literature on the number of seeds that should be used (this also depends on the specific MOEA that one adopts). In general, the idea is to use at least 10 seeds, so as to contribute diversity to a given MOEA's search results. Naturally, the number of seeds that one can use depends on the computational requirements of the simulation model as well as the available computational power.

To measure the convergence of the optimization algorithm, we used the hypervolume indicator, which captures both convergence and diversity of a Pareto front (see Reed et al., (2013)). Results indicated that the algorithm reached convergence after ∼150 iterations.

R: Page 14, lines 14-17: I disagree this is a real limitation, since proposing a universal rule system for reservoir operations is not an aim of this paper. Instead, a consequence of your work is that hydrological model calibration with reservoirs and a bespoke release rule means that the model still captures key hydrological processes once the release rules change (because there are new reservoirs / because reservoirs' purposes evolve). In contrast, if reservoirs are not represented to begin with, once the number and / or operations of reservoirs evolves, the model's hydrological parameters have to be recalibrated every time.

A: Thanks for this comment. We indeed said that this is not a limitation (see line 15), but we agree that the sentence was structured in an awkward–and potentially misleading–way. We will proceed by modifying this part of the Conclusions, so as to better emphasize the importance of a correct representation of water reservoirs in large-scale hydrological models.

References

He, D., Lu, Y., Li, Z., Li, S. (2009). Watercourse environmental change in Upper Mekong. The Mekong. Academic Press, 335-362.

Reed, P.M., Hadka, D., Herman, J.D., Kasprzyk, J.R., Kollat, J.B. (2013). Evolutionary multiobjective optimization in water resources: The past, present, and future. Advances in Water Resources, 51, 438-456.

Tang, X., Zhang, J., Wang, G., Yang, Q., Yang, Y., Guan, T., Liu, C., Jin, J., Liu, Y., Bao, Z. (2019). Evaluating Suitability of Multiple Precipitation Products for the Lancang River Basin. Chinese Geographical Science, 29(1), 37-57.

Wang, Z., Chen, J., Lai, C., Zhong, R., Chen, X., Yu, H. (2018). Hydrologic assessment of the TMPA 3B42-V7 product in a typical alpine and gorge region: the Lancang River basin, China. Hydrology Research, 49(6), 2002-2015.

[Figure]

**Fig. 1.** Observed versus modelled streamflow time series at Jiuzhou station.

[Figure]

**Fig. 2.** Average annual values of simulated baseflow (a,b) and runoff (c,d) simulated by the selected models (with and without reservoir) during the period 1996-2005.

---

## Author Response (AR1)

SINGAPORE UNIVERSITY OF
TECHNOLOGY AND DESIGN

Reply to reviewers of paper hess-2019-334

**On the representation of water reservoir storage and operations in large-scale hydrological models: implications on model parameterization and climate change impact assessments**

Thanh Duc Dang, AFM Kamal Chowdhury, Stefano Galelli

Resilient Water Systems Group
Pillar of Engineering Systems and Design
SUTD, 8 Somapah Road
Singapore 487372
T. +65 6303 6600
http://people.sutd.edu.sg/~stefano_galelli/

**Editor**

Dear authors,

your contribution seems to fit well into the scope of HESS and the topic is timely and well presented. The two reviewers have suggested some improvements and I strongly encourage you to follow those suggestions carefully and try to improve your work accordingly. Otherwise, please explain why you disagree with some suggestion of the reviewers:

*We would like to thank the Editor for the positive response and opportunity to revise our work. Following the reviewers' suggestions, we:*

- *Included a sensitivity analysis of the model (VIC) output with respect to the six soil parameters used in the calibration process. This analysis (described in the Supplement) helped us strengthen the results regarding the parameterization of models with / without reservoirs (Section 4.1);*
- *Used data from an upstream gauging station to (1) validate the selected models and (2) further study the impact of a flawed parameterization on climate change impact assessment. These new results are included in the revised version of the manuscript (Figures 9 and 11);*
- *Clarified a few minor aspects in Sections 2, 3, and 4.*

*Finally, please note that in our reply-to-reviewers line numbers correspond to the marked-up version of the manuscript.*

I have an additional request: You state that you can calibrate your model satisfactorily, both with and without reservoirs. Of course, this is somehow surprising. For instance, I do not think that one can model the Yangtze River downstream the 3 Gorges Dam without explicitly taking care of this (huge) reservoir. Thus, your findings might be dependent on the size and total storage volume of the reservoirs. Please elaborate on that. In addition, your model-parameters with considering reservoirs should result in more realistic soil parameters, I think. Can you also comment on that?

*Thanks for raising these two points, on which we elaborated in Section 5. Please refer to line 29-34 (page 15) and 4-8 (page 16, marked-up manuscript).*

**Reply to reviewer #1**

**General Comments:**

This paper presents a computational framework based on the Variable Infiltration Capacity (VIC) model and a Multi-Objective Evolutionary Algorithm that enables to analyze the effects of water reservoir representation on the parameterization of hydrological models. The modelling approach was applied to the upper Mekong river basin, upstream the Chiang Saen gauging station, considering two configuration schemes with and without water reservoirs. The authors exposed the theme in a clear, logical sequence, which resulted in a well-written comprehensive text. In my opinion, this work should be accepted for publication at the Hydrology and Earth System Sciences (HESS) journal after minor revisions. Attached you find some comments/suggestions.

*We thank the reviewer for the positive comments and practical suggestions for improving the manuscript.*

Introduction: A comprehensive literature review was done concerning hydrological modelling applied to large river basins, highlighting the limited number of model approaches that enables the direct representation of reservoir water storage with target operating rules to achieve pre-defined objectives. Furthermore, the scientific contribution of this manuscript is clearly defined from line 21 to 34 (Page 03).

*Thanks for the positive feedback.*

Study area: The study area description looks poor. Please include more information concerning soil properties and classes, land use, geology and geomorphology, just to enable a better comprehension of main driving forces related to hydrological processes in the Upper Mekong river basin.

*We included more details regarding soil properties, land use, geology and geo-morphology in Section 2. Please refer to line 32-33 (page 4) and 1-3 (page 5) of the marked-up manuscript.*

In Section 3.2.1 you present the way the input variables (Land use and land cover data, soil data, and flow direction) are obtained/processed, but a discussion about those data is lacking.

*We included a discussion about these data in the second paragraph of Section 3.2.1. Please refer to line 1-5 (page 8) of the marked-up manuscript.*

Materials and methods: The way used to achieve the objectives of the manuscript is clearly presented and well-organized. Nevertheless, I would suggest including a sensitivity analysis of the model results to the parameters controlling the rainfall-runoff process ($D_s$, $D_{max}$, $W_s$, $b$, $d_1$ and $d_2$) in the Variable Infiltration Capacity model.

*Thanks for the suggestion. We indeed explored the sensitivity of the model output with respect to the values of these six parameters (for both model configurations). Specifically, we linked VIC (with and without reservoirs) with the SAFE toolbox (Pianosi et al., 2015), which provides a number of global sensitivity algorithms— among which we chose the extended Fourier Amplitude Sensitivity Test, or eFAST.*

*The results of the sensitivity analysis show that all parameters are indeed important with respect to the model output. The only exception is the depth of the second soil layer (d2), which does not appear to depend on the absence (or presence) of water reservoirs (see the description of Figure 6 provided in Section 4.1). Since the description of the sensitivity analysis is a bit unwieldy, we preferred to keep it in the Supplement, to which we refer in Section 4.1.*

Results: The questions raised in the introduction section were answered. The results showed that a flawed model parametrization by disregarding anthropogenic interventions (such as hydraulic infrastructure) led to the overestimation of baseflow and runoff during the dry season to compensate water release related to hydropower production. Despite this, I would suggest discussing more deeply the model parameterization and results. According to the authors, the VIC model has been previously used by other researchers. Could your results be compared with them? Or other models with similar purpose.

*Literature offers a few studies that implemented VIC model to the Mekong River basin (e.g., Zhong et al. (2019), Chang et al. (2019)). Unfortunately, these studies do not report details about the parameterization—they tend to focus on the modelling accuracy. The same comment applies to studies that applied other hydrological models to the Mekong (e.g., Lauri et al. (2012), Rasanen et al. (2012), Hoang et al. (2016)). For this reason, we could not carry out a direct comparison between our parameterization and the one obtained by previous studies.*

**Specific Comments:**

1) Page 04, line 07: How large is the Upper Mekong river basin? Please include the catchment area in km2.

*The Upper Mekong river basin has an area of about 167,400 $km^2$; that's about 24% of the Mekong's basin area. We included this information in Section 2 (Study area).*

2) Page 04, lines 07-08: The information of elevation ranging from 362 m to 6,494 m should be included.

*We added this information.*

3) Page 04, line 30: Please include a complementary information explaining why the hydrological alterations became more evident since 1992. The largest reservoir (Xi'er He 1) was built before in 1989.

*Previous works focusing on the hydrological alterations caused by hydropower dams in the Lancang basin generally consider two periods, namely pre and post 1991 (e.g.,*

*1960-1991 and 1992-2013) (Cochrane et al. (2014), Lu et al. (2014), Dang et al. (2016)). The reason for this choice is due to Manwan dam, which received significant public attention because of its location—it is indeed the first dam built on the main stem of the river. We understand that our sentence is potentially misleading, so we clarified this point in the revised version of the manuscript (please refer to line 21-23, page 4, marked-up manuscript).*

4) Page 05, line 19 and Table 1: How the feasible ranges of the two soil layers thickness (d1 and d2) were defined?

*The definition of the feasible ranges is based on the ranges that are adopted in other studies applying VIC model to large catchments (e.g., Dan et al. (2012), Park and Markus (2014), Xue et al. (2015)). Note that the same ranges are also recommended by Wi et al. (2017), who developed a general-purpose user-friendly software package for VIC. We clarified this point in the caption of Table 1.*

5) Page 06, line 29: How the target water level is defined for each reservoir?

*The target water level is defined as follows. First, we determine the minimum and maximum water levels that a reservoir should reach within a year. In our case, we use the minimum and maximum elevation levels, which are given in the design specifications of each reservoir. Then, we set the time at which the minimum and maximum water levels should be reached. In our case, we use the months of May and November, which correspond to the beginning and end of the wet season. Finally, we connect these points with a piecewise linear function that gives us the daily target level for each calendar day. We further clarified this point in the second paragraph of Section 3.1.2.*

6) Page 08 lines 02-17: What processes are considered in the reservoir water balance and operation? Are infiltration loss and groundwater inflow disregarded?

*We considered three main processes, namely inflow, release, and evaporation. As noted by the reviewer, there are two other processes that could be considered, that is, infiltration and seepage (via dam body, abutment, and foundation). The reason for which we disregarded them is twofold. First, the Upper Mekong basin is a mountainous region, with mostly rocks and a shallow Quaternary alluvium (Gupta, 2009; Carling, 2009), so the infiltration losses are to some extent marginal as compared to inflow, release, and evaporation. Second, the dams considered in our study are built with concrete (and with rocky abutments and foundations), so seepage is indeed limited. We touched upon this point in Section 3.2.2 (last paragraph).*

7) Page 09, line 11: "parellelized" should be changed to "parallelized".

*Thanks for spotting this typo.*

8) Page 11, lines 05-24: Please discuss the pattern observed in Figure 6, with different ranges of the model parameters for the simulation with and without reservoirs. Some parameters presented a more spread pattern for the scenario with reservoirs and a more uniform one for the scenario without reservoir and vice versa.

*Thanks for the suggestion. We discuss about this pattern in both Section 4.1 (line 7-9, page 12, marked-up manuscript) and Supplement.*

9) Fig. 2: Please use the term "modelling approach" instead "model" (Figure legend).

*We think that the term "model" is more appropriate, since the figure depicts two specific numerical models (i.e., VIC's rainfall-runoff and routing modules) and an optimization algorithm (i.e., epsilon-NSGA-II)—and not a family of similar models that could be denoted with the term "approach".*

10) Fig. 3: Please include a legend describing the four levels highlighted in Figure 3c.

*Modified as suggested*

11) Fig 8: I would suggest to reduce the number of baseflow intervals to enable a better visualization of model results related to the configuration scheme with and without reservoirs.

*Thanks for the suggestion. We indeed modified the number of baseflow and runoff intervals.*

**Reply to reviewer #2**

**General Comments:**

This work contributes to literature on the representation of human interactions in hydrological model by demonstrating the importance of modeling reservoirs. Using the high-profile case of the Lancang / Mekong river basin, they calibrate a large-scale hydrological model (VIC), both with and without reservoirs. They show that while there exist parameter sets for which the model can be calibrated in both cases, the without- reservoir model compensate the absence of human-operated storage by artificially creating soil storage. Then they go on to investigate the reactions of both models to climate change.

This is an interesting, timely and well-written piece of work that fits well within the scope of HESS. My comments are mainly at how the results are handled and interpreted.

*We thank the reviewer for the positive feedback.*

1) While authors explain very well the consequences of the no-reservoir calibration in terms of structural model behavior, they do not show what that means for water resources appraisals. Indeed, both models are calibrated to have the same behavior downstream of 1 gauge. The crucial difference is that the model including reservoirs can be 1) validated at other (upstream) sites, and 2) validated with post 2005 dams (or the results can be extrapolated with new dams or new operating rules). This point is not really demonstrated.

*Thanks for this suggestion. We totally agree that we can strengthen our message by implementing one of these two validations. In particular, we believe that the first option is the most feasible, although it presents some minor challenges on which we elaborate below. (Our thoughts regarding the second option are outlined as a response to comment no. 4.)*

*Discharge data in the Chinese section of the Mekong basin are not available to the international research community. More specifically, only water level data for the flood season (June-October) at Yunjinghong station (located downstream of Jinghong reservoir; see Figure 1 in the manuscript) are available through the Mekong River Commission (http://www.mrcmekong.org/news-and-events/news/mrc-and-china-renew-pact-on-water-data-provision-and-other-cooperation-initiatives/). Therefore, we opted for retrieving data from three previous studies (He et al. (2009), Wang et al. (2018), and Tang et al. (2019)) that published monthly discharge values at Jiuzhou station (located near Gonguoqiao reservoir; see Figure 1) for the period 1996-2005. When retrieving (digitalizing) the data, we found a maximum discrepancy between the three time series of about 10 $m^3/s$; a negligible value if we consider that the minimum discharge is roughly 400 $m^3/s$.*

*With this 10-year time series, we carried out a thorough validation exercise, whose results are illustrated in the figure below. We can note that the model calibrated without reservoirs largely overestimates the dry season flow and slightly*

*underestimates the wet season flow. This point is further discussed in Section 4.1 (line 27-31, page 12, marked-up manuscript).*

[Figure]

*Figure 1. Comparison between observed and simulated monthly discharges at Jiuzhou station over the period 1996–2005. Simulated data are produced by the two selected models with and without reservoirs (blue and red dots, respectively).*

2) Likewise, I disagree that the results from climate models are very different. Yes, there are some small difference, and authors do a good job of explaining them, but these differences are arguably small compared with the uncertainty surrounding downscaled climate projections. In fact, it is remarkable how robust the no-reservoir model results are with climate change.

*We agree with the reviewer that the uncertainty surrounding the climate change projections is larger than the difference (in discharge) due to the representation of water reservoirs in VIC. This said, we also note that such difference is non-negligible and consistent across both RCPs. We thus proceeded by slightly toning down this result in the revised version of the manuscript (please refer to line 28-34, page 13, marked-up manuscript).*

3) I would advise authors to try and show how both calibrations differ at upstream gauges (where only the with-reservoir calibration would yield sensible results) where there is data, then carry on that comparison with climate change.

*Yes, we included an additional validation based on data at an upstream gauge, as explained in our reply to comment no. 1. We also included the same comparison under the climate change scenarios (please refer to line 16-22, page 14, marked-up manuscript).*

4) Alternatively (but this is probably more work), authors can focus on post-2005 years with the addition of new dams and look at the differences between both model at the outlet. And again look at how the two models differ with climate change then.

*The validation based on discharge data collected in the upstream reaches of the basin is indeed simpler to implement, so we preferred to adopt such option. This said, we*

*understand the potential of this second suggested validation, so we did elaborate on this point in the concluding remarks.*

5) Figure 8 is not very clear and could be replaced with curves comparing average monthly basinwide runoff for the two calibrations (and same for baseflow). This would show explicitly how the no-reservoir calibration compensates for the reservoirs. Alternatively, authors could map the dry / wet season cell-by-cell differences in baseflow / runoff between the two calibrations.

*Yes, we agree that Figure 8 is not that clear. The option of replacing it with curves comparing average monthly basin-wide runoff for the two calibrations is sound. Yet, such figure would not represent explicitly the spatial variability or runoff and baseflow; something that we deem important. We thus preferred to keep the current 'skeleton' of Figure 8, which we improved by modifying the number of baseflow intervals (as suggested by reviewer #1).*

6) Figures 9 and 10 don't bring much and could be relegated to supplementary material to make place for new figures that can show the consequences of omitting reservoirs, e.g., as suggested in 3) and 4)

*Yes, we moved both Figure 9 and 10 to the Supplement.*

**Detailed Comments:**

Figure 1.b: the scale, useful in 1.a, is missing.

*Modified as suggested.*

Section 3.1.2: a short description of the version 4.2 of VIC's routing module would be helpful here. Recall that this research is useful for all large-scale hydrological models, not just for VIC experts.

*We provided a brief description of VIC's routing module. However, we thought it's better to include such description in Section 3.1.1, which focuses entirely on VIC (section 3.1.2 is about the modification we introduced). Please refer to line 21-25 (page 5, marked-up manuscript).*

Page 8, lines 2-9: there is a recent publication presenting a database of reservoir storage-area-depth relationships in Yigzaw et al (2018) there https://agupubs.onlinelibrary.wiley.com/doi/10.1029/2017WR022040 could that affect your results in significant ways?

*Thanks for pointing us to this paper. We noticed that only Manwan reservoir is in this new database, so we believe this cannot impact our model implementation / results. This said, we believe that the adoption of Liebe's method should provide robust results—the method has been tested and adopted by several state-of-the-art studies in large-scale hydrology.*

Page 9, lines 8-12: how many parallel processors are running? (l. 12) by runtime, do you mean the wall clock time or the total computational time used by all the processors (i.e. wallclock time times # of processors)? How do we know that 20 seeds are enough? And that the algorithm has converged after 250 function evaluations? Finally, what is the runtime for one run of VIC for that basin?

*For each (of the 20) seed(s), we used four cores. As for the runtime, we reported the wall-clock time per core. The runtime of VIC for the basin is about 50 mins (on one core). We clarified these details in the revised version of the manuscript (line 29-30, page 9, marked-up manuscript).*

*There are no specific guidelines in the literature on the number of seeds that should be used (this also depends on the specific MOEA that one adopts). In general, the idea is to use at least 10 seeds, so as to contribute diversity to a given MOEA's search results. Naturally, the number of seeds that one can use depends on the computational requirements of the simulation model as well as the available computational power.*

*To measure the convergence of the optimization algorithm, we used the hypervolume indicator, which captures both convergence and diversity of a Pareto front (see Reed et al., (2013)). Results indicated that the algorithm reached convergence after ~150 iterations.*

Page 14, lines 14-17: I disagree this is a real limitation, since proposing a universal rule system for reservoir operations is not an aim of this paper. Instead, a consequence of your work is that hydrological model calibration with reservoirs and a bespoke release rule means that the model still captures key hydrological processes once the release rules change (because there are new reservoirs / because reservoirs' purposes evolve). In contrast, if reservoirs are not represented to begin with, once the number and / or operations of reservoirs evolves, the model's hydrological parameters have to be recalibrated every time.

*Thanks for this comment. We agree that the sentence was structured in an awkward—and potentially misleading—way, so we removed it. We also elaborated on the reviewer's comment in the latter part of the Conclusions (line 4-12, page 16, marked-up manuscript).*

[revised manuscript text omitted]

---

## Author Response (AR3)

SINGAPORE UNIVERSITY OF
TECHNOLOGY AND DESIGN

Reply to the editor of paper hess-2019-334

**On the representation of water reservoir storage and operations in large-scale hydrological models: implications on model parameterization and climate change impact assessments**

Thanh Duc Dang, AFM Kamal Chowdhury, Stefano Galelli

Resilient Water Systems Group
Pillar of Engineering Systems and Design
SUTD, 8 Somapah Road
Singapore 487372
T. +65 6303 6600
http://people.sutd.edu.sg/~stefano_galelli/

**Editor**

Dear colleagues,

your paper is rather ready for publication. However, as the editor in charge for this manuscript, I suggest you add a better understandable description of your innovative part of the VIC-model (Water reservoir storage and operations, section 3.1.2). This means, you should give the equations which you apply in the routine (currently you only give text), and you also may add a graphical scheme which enables the reader to grasp the main components of this new routine.

I think this should be feasible.
Kind regards
Axel Bronstert

*Dear Professor Bronstert,*

*We would like to thank you for the positive response and opportunity to revise our work. Following your suggestions, we:*

- *Introduced the key equations characterizing VIC-Res (i.e., reservoir mass balance and release function) in Section 3.1.2;*
- *Added a graphical scheme that illustrates how the information on water reservoirs is implemented in the model. To keep the paper focused on the model (rather than the implementation), we preferred to add the scheme to the supplement—please refer to Section S1.*

*Best,*
*Stefano Galelli (corresponding author)*